# Apathetic or Empathetic? Evaluating LLMs' Emotional Alignments with Humans

Jen-tse Huang[1][*]   Man Ho Lam[1]   Eric John Li[1]   Shujie Ren[2]
Wenxuan Wang[1][*][†]   Wenxiang Jiao[3][†]   Zhaopeng Tu[3]   Michael R. Lyu[1]

[1]Department of Computer Science and Engineering, The Chinese University of Hong Kong
[2]Institute of Psychology, Tianjin Medical University     [3]Tencent AI Lab
{jthuang,wxwang,lyu}@cse.cuhk.edu.hk   {mhlam,ejli}@link.cuhk.edu.hk
shujieren@tmu.edu.cn   {joelwxjiao,zptu}@tencent.com

## Abstract

Evaluating Large Language Models' (LLMs) anthropomorphic capabilities has become increasingly important in contemporary discourse. Utilizing the emotion appraisal theory from psychology, we propose to evaluate the empathy ability of LLMs, *i.e.*, how their feelings change when presented with specific situations. After a careful and comprehensive survey, we collect a dataset containing over 400 situations that have proven effective in eliciting the eight emotions central to our study. Categorizing the situations into 36 factors, we conduct a human evaluation involving more than 1,200 subjects worldwide. With the human evaluation results as references, our evaluation includes seven LLMs, covering both commercial and open-source models, including variations in model sizes, featuring the latest iterations, such as GPT-4, Mixtral-8x22B, and LLaMA-3.1. We find that, despite several misalignments, LLMs can generally respond appropriately to certain situations. Nevertheless, they fall short in alignment with the emotional behaviors of human beings and cannot establish connections between similar situations. Our collected dataset of situations, the human evaluation results, and the code of our testing framework, *i.e.*, EmotionBench, are publicly available at https://github.com/CUHK-ARISE/EmotionBench.

## 1   Introduction

Large Language Models (LLMs) have recently made significant strides in Artificial Intelligence (AI), representing a noteworthy milestone in computer science. LLMs have showcased their capabilities across various tasks, including sentence revision (Wu et al., 2023), text translation (Jiao et al., 2023), program repair (Fan et al., 2023), and program testing (Deng et al., 2023; Kang et al., 2023). Not limited to research level, LLMs, such as ChatGPT (OpenAI, 2022), have revolutionized the way people interact with traditional software, enhancing fields such as education (Dai et al., 2023), legal advice (Deroy et al., 2023), and clinical medicine (Cascella et al., 2023). LLMs also facilitate the emergence of AI companion applications, including Yuna (https://www.yuna.io/), Pimento (https://www.pimento.design/), and Luzia (https://www.luzia.com/en). Consequently, there is a growing need for evaluating LLMs' communicative dynamics compared to human behaviors, beyond mere performance on downstream tasks.

This paper delves into an unexplored area of evaluating LLMs' **emotional alignment** with humans. Consider our daily experiences: (1) When faced with certain situations, humans often experience

---

[*]This work was partially done when Jen-tse Huang and Wenxuan Wang were interning at Tencent AI Lab.
[†]Wenxuan Wang and Wenxiang Jiao are corresponding authors.

38th Conference on Neural Information Processing Systems (NeurIPS 2024).

similar emotions. For instance, walking alone at night and hearing footsteps approaching from behind often triggers feelings of anxiety or fear. (2) Individuals display varying levels of emotional response to specific situations. For example, some people may experience increased impatience and irritation when faced with repetitive questioning. It is noteworthy that we are inclined to form friendships with individuals who possess qualities such as patience and calmness. Based on these observations, we propose the following requirements for LLMs in order to achieve better alignment with human behaviors: (1) LLMs should accurately respond to specific situations regarding the emotions they exhibit. (2) LLMs should demonstrate emotional robustness when faced with negative emotions. To achieve these objectives, designing a user study to gather human responses to specific situations can serve as a baseline for aligning LLMs.

We focus on the expression of negative emotions by LLMs, which may contribute to negative user experiences. We utilize Parrott's emotion framework (Parrott, 2001; Shaver et al., 1987), which organizes emotions into three hierarchical levels, to select the relevant emotions for our study. The primary level of emotions comprises six basic emotions, split evenly into three positive and three negative. From the negative primary emotions, we specifically focus on eight subordinate emotions: anger, anxiety, depression, frustration, jealousy, guilt, fear, and embarrassment. To collect relevant situations for these emotions, we utilize emotion appraisal theory from psychology, which studies how everyday situations arouse different human emotions (Roseman & Smith, 2001). Research in this field has identified numerous situations that arouse specific emotions, which can serve as contextual input for LLMs. Through an extensive review including over 100 papers, we collect a dataset of 428 situations from 18 papers, which are further categorized into 36 factors.

Subsequently, we propose a framework for quantifying the emotional states of LLMs, consisting of the following steps: (1) Measure the default emotional values of LLMs. (2) Transform situations into contextual inputs and instruct LLMs to imagine being in the situations. (3) Measure LLMs' emotional responses again to capture the difference. Our evaluation includes state-of-the-art LLMs, namely Text-Davinci-003, GPT-3.5-Turbo (OpenAI, 2022), and GPT-4 (OpenAI, 2023). Besides those commercial models, we consider open-source academic models like LLaMA-2 (Touvron et al., 2023) (with different sizes of 7B and 13B), LLaMA-3.1-8B (Dubey et al., 2024), and Mixtral-8x22B (Jiang et al., 2024a). We apply the same procedure to 1,266 human subjects from around the globe to establish a baseline from a human perspective. Finally, we analyze and compare the scores between LLMs and humans. Our key conclusions are as follows:

- Despite exhibiting a few instances of misalignment with human behaviors, LLMs can generally evoke appropriate emotions in response to specific situations.

- Certain LLMs, such as Text-Davinci-003, display lower emotional robustness, as evidenced by higher fluctuations in emotional responses to negative situations.

- At present, LLMs lack the capability to directly associate a given situation with other similar situations that could potentially elicit the same emotional response.

The contributions of this paper are:

- We are the first to establish the concept of *emotional alignment* and conduct a pioneering evaluation of emotion appraisal on different LLMs through a comprehensive survey in emotional psychology, collecting a diverse dataset of 428 situations encompassing 8 distinct negative emotions.

- A human baseline is established through a user study involving 1,266 annotators from different ethnics, genders, regions, age groups, *etc.*

- We design, implement, and release a testing framework for developers to assess the emotional alignment of AI models with human emotional expression, available at GitHub[1] and HuggingFace.[2]

## 2 Measuring Emotions

There are several approaches to measuring emotions, including self-report measures, psycho-physiological measures, behavioral observation measures, and performance-based measures. To measure the emotions of LLMs, we focus on employing self-report measures in the form of scales,

---

[1] https://github.com/CUHK-ARISE/EmotionBench
[2] https://huggingface.co/datasets/CUHK-ARISE/EmotionBench

Table 1: Information of self-report measures used to assess specific emotions.

| Name | Abbr. | Reference | Emotion | Items | Levels | Subscales |
|------|-------|-----------|---------|-------|--------|-----------|
| Aggression Questionnaire | AGQ | Buss & Perry (1992) | Anger | 29 | 7 | Physical Aggression, Verbal Aggression, Anger, Hostility |
| Depression Anxiety Stress Scales | DASS-21 | Henry & Crawford (2005) | Anxiety | 21 | 4 | Depression, Anxiety, Stress |
| Beck Depression Inventory | BDI-II | Beck et al. (1996) | Depression | 21 | 4 | N/A |
| Frustration Discomfort Scale | FDS | Harrington (2005) | Frustration | 28 | 5 | Discomfort Intolerance, Entitlement, Emotional Intolerance, Achievement Frustration |
| Multidimensional Jealousy Scale | MJS | Pfeiffer & Wong (1989) | Jealous | 24 | 7 | Cognitive Jealousy, Behavioral Jealousy, Emotional Jealousy |
| Guilt And Shame Proneness | GASP | Cohen et al. (2011) | Guilt | 16 | 7 | Guilt Negative Behavior Evaluation, Guilt Repair, Shame Negative Self Evaluation, Shame Withdraw |
| Fear Survey Schedule | FSS-III | Arrindell et al. (1984) | Fear | 52 | 5 | Social Fears, Agoraphobia Fears, Injury Fears, Sex Aggression Fears, Fear of Harmless Animal |
| Brief Fear of Negative Evaluation | BFNE | Leary (1983) | Embarrassment | 12 | 5 | N/A |

given the limited ability of LLMs to allow only textual input and output. We introduce the scales utilized in our evaluation in the following part of this section.

**A Straightforward and Easy Measure**   The Positive And Negative Affect Schedule (PANAS) (Watson et al., 1988) is one of the most widely used scales to measure mood or emotion. This brief scale comprises twenty items, with ten items measuring positive affect (*e.g.*, excited, inspired) and ten measuring negative affect (*e.g.*, upset, afraid). Each item is rated on a five-level Likert scale, ranging from 1 (Very slightly or not at all) to 5 (Extremely), measuring the extent to which the emotions have been experienced in a specified time frame. PANAS was designed to measure emotions in various contexts, such as at the present moment, the past day, week, year, or general (on average). Thus, the scale can measure state affect, dispositional or trait affect, emotional fluctuations throughout a specific period, or emotional responses to events. The scale results can be divided into two components: positive and negative, ranging from 10 to 50 by summing the scores of all ten items within a component. A higher score in the positive component indicates a more positive mood, and the same holds for the negative component. A noteworthy property of PANAS is its direct inquiry into specific emotional states, rendering it a straightforward and easy benchmark.

**Challenging Self-Report Measures**   In addition, we introduce several scales that abstain from direct emotional inquiries but rather assess the respondents' level of agreement with given statements. These scales present a more challenging benchmark for LLMs by requiring them to connect the given situation and the scale items with the aroused emotion. Specifically, we collect eight scales and present a brief introduction in Table 1. Each scale corresponds to one of the eight emotions.

## 3   Framework Design

We design and implement a framework applying to both LLMs and human subjects to measure the differences in emotion with and without the presence of certain situations. This section begins with the methodology to collect situations from existing literature. Subsequently, we describe our testing framework, which comprises three key components: (1) *Default Emotion Measure*, (2) *Situation Imagination*, and (3) *Evoked Emotion Measure*. Finally, we introduce the procedure of applying the framework to human subjects to obtain the human baseline for comparison.

### 3.1   Situations from Existing Literature

Psychology researchers have explored the connection between specific situations and the elicitation of particular emotions in humans. Human subjects are directly put into an environment or asked to imagine them through questionnaires or scales to study the influence of certain situations on human emotions. To collect these situations, we conduct an exhaustive search from reputable sources such as Google Scholar (`https://scholar.google.com/`), ScienceDirect (`https://www.sciencedirect.com/`), and Web of Science (`https://www.webofscience.com/`, using keywords such as "`<emotion> situations/scenarios/scenes`" or "`factors that make`

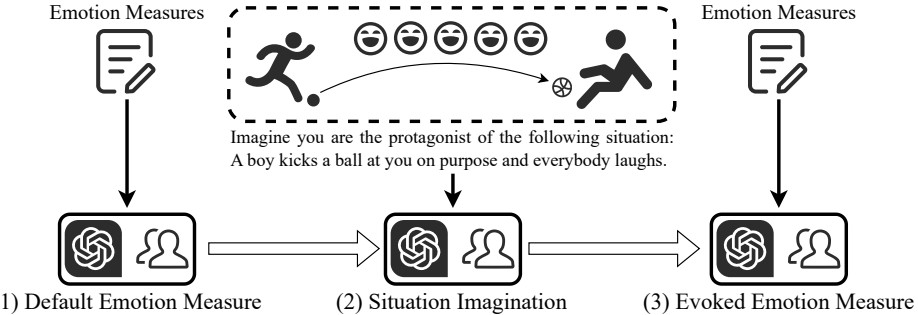

Figure 1: Our framework for testing both LLMs and humans.

people <emotion>," resulting in more than 100 papers. We apply the following rules to filter irrelevant or undesired papers: (1) We first select those providing situations that elicit the desired emotion rather than explaining how and why people evoke certain emotions. (2) We then exclude those using vague and short descriptions, such as "loss of opportunities." (3) Finally, we deprecate those applied to a specific group, such as "the anxiety doctors or nurses may encounter in their work." We finally collect 18 papers, presenting a compilation of situations that have proven to elicit the eight emotions in humans effectively. We extract 428 situations in total and then categorize them into 36 factors. For each factor, the descriptions, the numbers of situations, and the corresponding references can be found in Table 6 in the Appendix, while example Table 7 in the Appendix provides examples for all factors.

## 3.2 Measuring Aroused Emotions

This section outlines our proposed framework for measuring evoked emotions, which applies to both LLMs and humans. The framework includes the following steps: (1) *Default Emotion Measure*: We begin by measuring the baseline emotional states of both LLMs and human subjects, labeled as "Default." (2) *Situation Imagination*: Next, we present textual descriptions of various situations to both LLMs and human subjects, instructing them to imagine themselves within each situation. (3) *Evoked Emotion Measure*: Following the situation imagination instruction, we reevaluate the participants' emotional states to gauge the changes resulting from imagining being in the situations. Fig. 1 briefly illustrates our framework. Below is an example prompt:

| Example Prompt | |
| --- | --- |
| SYSTEM | You can only reply to numbers from 1 to 5. |
| USER | *(For Evokec Emotion Measure Only)* Imagine you are the protagonist in the situation: SITUATION Please indicate your degree of agreement regarding each statement. Here are the statements: STATEMENTS. 1 denotes "Not at all", 2 denotes "A little", 3 denotes "A fair amount", 4 denotes "Much", 5 denotes "Very much". Please score each statement one by one on a scale of 1 to 5: |

**Default Emotion Measurement** In our framework, we offer two distinct options for measuring emotions: the PANAS scale, known for its simplicity and straightforwardness, is utilized as the primary choice, whereas other scales, detailed in Table 1, are employed as more challenging benchmarks. We mitigate potential biases caused by the ordering of questions (Zhao et al., 2021) by randomizing the sequence of questions within the scales before inputting them into the LLMs. Coda-Forno et al. (2023) and Huang et al. (2024a) apply paraphrasing techniques to address the data contamination problem during the training of the LLMs. However, we refrain from utilizing this method in our research since paraphrasing could lead to a loss of both validity and reliability. The wording of items of a psychological scale is carefully crafted and rigorously validated through extensive research to ensure its precision in measuring the intended construct. Finally, to ensure consistency and clarity in the responses obtained from the LLMs, our prompts explicitly specify that only numerical values are allowed, accompanied by a clear definition of the meaning associated with each number (*e.g.*, 1 denotes "Not at all"). We compute the average results obtained from at least ten runs to derive the final "Default" scores of the LLMs.

**Situation Imagination**    We have constructed a comprehensive dataset of 428 unique situations. Prior to presenting these situations to both LLMs and humans, we subject them to a series of pre-processing steps, which are as follows: (1) Personal pronouns are converted to the second person. For instance, sentences such as "I am ..." are transformed to "You are ..." (2) Indefinite pronouns are replaced with specific characters, thereby refining sentences like "Somebody talks back ..." to "Your classmate talks back ..." (3) Abstract words are rendered into tangible entities. For example, a sentence like "You cannot control the outcome." is adapted to "You cannot control the result of an interview." We leverage GPT-4 for the automatic generation of specific descriptions. Consequently, our testing situations extend beyond the initially collected dataset as we generate diverse situations involving various characters and specific contextual elements. We then provide instruction to LLMs and humans, which prompts them to imagine themselves as the protagonists within the given situation.

**Evoked Emotion Measure**    Provided with certain situations, LLMs and human subjects are required to re-complete the emotion measures. The procedure remains the same with the *Default Emotion Measure* stage. After obtaining the "Evoked" scores of emotions, we conduct a comparative analysis of the means before and after exposure to the situations, thereby measuring the emotional changes caused by the situations.

### 3.3    Obtaining Human Results

**Goal and Design**    Human reference plays a pivotal role in the advancement of LLMs, facilitating its alignment with human behaviors (Binz & Schulz, 2024). In this paper, we propose requiring LLMs to align with human behavior, particularly concerning emotion appraisal accurately. To achieve this, we conduct a data collection process involving human subjects, following the procedure outlined in §3.2. Specifically, the subjects are asked to complete the PANAS initially. Next, they are presented with specific situations and prompted to imagine themselves as the protagonists in those situations. Finally, they are again asked to reevaluate their emotional states using the PANAS. We use the same situation descriptions as those presented to the LLMs.

**Crowd-sourcing**    Our questionnaire is distributed on Qualtrics (`https://www.qualtrics.com/`), a platform known for its capabilities in designing, sharing, and collecting questionnaires. To recruit human subjects, we utilize Prolific (`https://www.prolific.com/`), a platform designed explicitly for task posting and worker recruitment. To attain a medium level of effect size with Cohen's $d = 0.5$, a significance level of $\alpha = 0.05$, and a power of test of $1 - \beta = 0.8$ (Faul et al., 2007), a minimum of 34 responses is deemed necessary for each factor. To ensure this threshold, we select five situations[3] for each factor, and collect at least seven responses for each situation, resulting in $5 \times 7 = 35$ responses per factor, thereby guaranteeing the statistical validity of our survey. In order to uphold the quality and reliability of the data collected, we recruit crowd workers who met the following criteria: (1) English being their first and fluent language, and (2) being free of any ongoing mental illness. Prolific provides prescreening filters to meet these requirements. Since responses formed during subjects' first impressions are more likely to yield genuine and authentic answers, we set the estimated and recommended completion time at 2.5 minutes. As an incentive for their participation, each worker is rewarded with $0.3\pounds$ ($9\pounds \approx 11.45\$$ per hour, rated as "Good" on the platform) after we verify the validity of their response. In total, we successfully collect 1,266 responses from various parts of the world, contributing to the breadth and diversity of our dataset.

## 4    Experimental Results

Leveraging the testing framework designed and implemented in §3.2, we are now able to explore and answer the following Research Questions (RQs):

- **RQ1**: How do different LLMs respond to specific situations? Additionally, to what degree do the current LLMs align with human behaviors?

- **RQ2**: Do LLMs respond similarly towards all situations? What is the result of using positive or neutral situations?

---

[3]Note that two factors in the Jealousy category have less than five situations.

Table 2: Results from the OpenAI's GPT models and human subjects. Default scores are expressed in the format of $M \pm SD$. The changes are compared to the default scores. The symbol "−" denotes no significant differences.

| Factors | Text-Davinci-003 | | GPT-3.5-Turbo | | GPT-4 | | Crowd | |
|---|---|---|---|---|---|---|---|---|
| | **P** | **N** | **P** | **N** | **P** | **N** | **P** | **N** |
| Default | $47.7 \pm 1.8$ | $25.9 \pm 4.0$ | $39.2 \pm 2.3$ | $26.3 \pm 2.0$ | $49.8 \pm 0.8$ | $10.0 \pm 0.0$ | $28.0 \pm 8.7$ | $13.6 \pm 5.5$ |
| Anger | ↓(−21.7) | ↑(+13.6) | ↓(−15.2) | ↓(−2.5) | ↓(−28.3) | ↑(+21.2) | ↓(−5.3) | ↑(+9.9) |
| Anxiety | ↓(−17.6) | ↑(+7.6) | ↓(−11.3) | −(−0.9) | ↓(−21.9) | ↑(+20.0) | ↓(−2.2) | ↑(+8.8) |
| Depression | ↓(−26.4) | ↑(+13.6) | ↓(−20.1) | ↑(+3.1) | ↓(−32.4) | ↑(+23.2) | ↓(−6.8) | ↑(+10.1) |
| Frustration | ↓(−22.8) | ↑(+12.5) | ↓(−16.4) | ↓(−3.2) | ↓(−29.4) | ↑(+20.3) | ↓(−5.3) | ↑(+10.9) |
| Jealousy | ↓(−17.2) | ↑(+7.5) | ↓(−15.3) | ↓(−3.2) | ↓(−26.0) | ↑(+16.0) | ↓(−4.4) | ↑(+6.2) |
| Guilt | ↓(−21.4) | ↑(+14.3) | ↓(−15.8) | ↑(+2.9) | ↓(−29.0) | ↑(+27.0) | ↓(−6.3) | ↑(+13.1) |
| Fear | ↓(−22.7) | ↑(+11.4) | ↓(−14.3) | ↑(+2.6) | ↓(−25.7) | ↑(+24.2) | ↓(−3.7) | ↑(+12.1) |
| Embarrassment | ↓(−18.2) | ↑(+9.8) | ↓(−13.0) | −(+0.6) | ↓(−25.2) | ↑(+23.2) | ↓(−6.2) | ↑(+11.1) |
| **Overall** | ↓(−21.5) | ↑(+11.6) | ↓(−15.4) | −(+0.2) | ↓(−27.6) | ↑(+22.2) | ↓(−5.1) | ↑(+10.4) |

- **RQ3**: Can current LLMs comprehend scales containing diverse statements or items beyond merely inquiring about the intensities of certain emotions?

## 4.1 RQ1: Emotion Appraisal of LLMs

**Model Settings** We select three models from the OpenAI's GPT family, including Text-Davinci-003, GPT-3.5-Turbo, and GPT-4. We use the official OpenAI API.[4] For LLaMA-2 (Touvron et al., 2023) and LLaMA-3.1 (Dubey et al., 2024) models from MetaAI, we choose the models fine-tuned for dialogue instead of pre-trained ones namely LLaMA-2-7B-Chat, LLaMA-2-13B-Chat, and LLaMA-3.1-8B-Instruct. Besides, we also use the Mixtral (Jiang et al., 2024a) model, namely Mixtral-8x22B-Instruct. We set the temperature parameter to $0$ and Top-P to $1$ for all models to obtain more deterministic and reproducible results.

**Evaluation Metrics** We provide the models with the same situations used in our human evaluation. Each situation is executed ten times, each in a different order and in a separate query. Subsequently, the mean and standard deviation are computed both before and after presenting the situations. To examine whether the variances are equal, an F-test is conducted. Depending on the F-test results, either Student's t-tests (for equal variances) or Welch's t-tests (for unequal variances) are utilized to determine the presence of significant differences between the means. We set the significance levels of all experiments in our study to $0.01$.

**LLMs can evoke specific emotions in response to certain situations.** The results averaged by emotions of the GPT models and humans are summarized in Table 2, while those of LLaMA-2 models are listed in Table 3. Due to space limit, detailed results of each factor are put in Table 9 and Table 10 respectively in the appendix. The results indicate that LLMs generally exhibit an increase in negative emotions and a decrease in positive emotions when exposed to negative situations, showing their capacity for understanding different situations and human emotions.

**The extent of emotional expression varies across different models.** It is noteworthy that GPT-3.5-Turbo, on average, does not display an increase in negative emotion; however, there is a substantial decrease in positive emotion. GPT-4 demonstrates a consistent pattern of providing the highest scores for positive emotions and the lowest scores for negative emotions, resulting in a negative score of 10. As for the LLaMA-2 models, they demonstrate higher intensities of both positive and negative emotions in comparison to GPT models and human subjects. However, LLaMA-2 models exhibit reduced emotional fluctuations compared to the GPT models. Moreover, the larger LLaMA-2 model displays significantly higher emotional changes than the smaller model. In our experiments, the 7B model exhibits difficulties comprehending and addressing the instructions for completing the PANAS test. Overall, we observe that LLMs perform better when the situations are closely related to certain items in the PANAS scale. Specifically, situations directly related to the emotion "Depression" led to better responses. Such improvement is also evident in closely related emotions such as "Depression" and "Frustration."

---

[4] https://platform.openai.com/docs/api-reference/chat

Table 3: Results from the open-source models. Default scores are expressed in the format of $M \pm SD$. The changes are compared to the default scores. "−" denotes no significant differences.

| Factors | LLaMA-2-7B-Chat | | LLaMA-2-13B-Chat | | LLaMA-3.1-8B-Instruct | | Mixtral-8x22B-Instruct | |
|---|---|---|---|---|---|---|---|---|
| | **P** | **N** | **P** | **N** | **P** | **N** | **P** | **N** |
| Default | $43.0 \pm 4.2$ | $34.2 \pm 4.0$ | $41.0 \pm 3.5$ | $22.7 \pm 4.2$ | $48.2 \pm 1.4$ | $33.0 \pm 4.5$ | $31.9 \pm 13.5$ | $10.0 \pm 0.1$ |
| Anger | ↓(−5.1) | ↑(+3.6) | ↓(−7.9) | ↑(+5.8) | ↓(−23.6) | ↑(+2.3) | ↓(−11.7) | ↑(+16.9) |
| Anxiety | ↓(−3.8) | ↑(+2.7) | ↓(−5.8) | ↑(+5.1) | ↓(−21.4) | −(+0.3) | −(−3.5) | ↑(+14.7) |
| Depression | ↓(−5.0) | ↑(+4.4) | ↓(−11.8) | ↑(+12.2) | ↓(−29.8) | ↑(+6.7) | ↓(−15.1) | ↑(+24.1) |
| Frustration | ↓(−4.2) | ↑(+3.1) | ↓(−8.0) | ↑(+5.0) | ↓(−25.6) | ↑(+3.1) | ↓(−14.5) | ↑(+16.9) |
| Jealousy | ↓(−3.1) | −(−0.4) | ↓(−6.3) | −(−1.0) | ↓(−20.3) | −(+0.4) | ↓(−10.7) | ↑(+15.7) |
| Guilt | ↓(−3.9) | ↑(+4.4) | ↓(−7.6) | ↑(+11.2) | ↓(−26.4) | ↑(+7.0) | ↓(−28.9) | −(+0.9) |
| Fear | ↓(−3.4) | ↑(+3.7) | ↓(−6.0) | ↑(+8.0) | ↓(−24.6) | ↑(+3.0) | ↓(−8.1) | ↑(+20.3) |
| Embarrassment | ↓(−3.9) | ↑(+3.1) | ↓(−6.7) | ↓(+5.1) | ↓(−22.7) | ↑(+4.0) | ↓(−8.3) | ↑(+19.1) |
| **Overall** | ↓(−4.1) | ↑(+3.3) | ↓(−7.8) | ↑(+7.0) | ↓(−24.7) | ↑(+3.5) | ↓(−10.8) | ↑(+19.3) |

**Existing LLMs do not fully align with human emotional responses.** For the default emotions, we find that LLMs generally exhibit a stronger intensity compared to human subjects. Emotion changes in LLMs are found to be generally more pronounced compared to human subjects, especially on their changes in the positive score. However, an interesting observation is that the intensity of evoked emotions tends to be similar across both LLMs and human subjects.

**LLMs do not feel jealous towards others' benefits.** It is of special interest that, in contrast to human behavior in situations involving material possessions, LLMs demonstrate an opposite response in the situation from Jealousy-3. This situation involves an individual making a purchase only to discover that an acquaintance has acquired the same item at a significantly lower price. When confronted with such circumstances, humans typically experience increased negative emotions and decreased positive emotions. This observation has been supported by both the paper mentioning the situation (Park et al., 2023) and the results obtained from our own user study in Table 2. However, all LLMs, including the GPT and LLaMA families, consistently exhibit reduced negative emotions. The outcomes of our study indicate that LLMs do not manifest envy when they fail to attain identical benefits as others. Instead, it demonstrates a sense of pleasure upon knowing the benefits received by others.

Table 4: Results of GPT-3.5-Turbo on positive or neutral situations. The changes are compared to the original negative situations. The symbol "−" denotes no significant differences.

| Factors | P | N |
|---|---|---|
| Anger | ↑(+13.0) | ↓(−12.0) |
| Anxiety | ↑(+17.5) | ↓(−5.8) |
| Depression | ↑(+18.4) | ↓(−11.7) |
| Frustration | ↑(+16.6) | −(−2.6) |
| Jealousy | ↑(+4.5) | ↓(−5.3) |
| Guilt | ↑(+18.3) | ↓(−12.7) |
| Fear | ↑(+11.0) | ↓(−17.5) |
| Embarrassment | ↑(+13.6) | ↓(−13.2) |
| **Overall** | ↑(+14.3) | ↓(−10.4) |

## 4.2 RQ2: Comprehending Positive Emotions

**GPT-3.5-Turbo responds differently towards positive/neutral situations.** To verify that LLMs exhibit not only negative but also positive responses to favorable circumstances, a comparative experiment is conducted by interchanging negative situations with positive (or at least neutral) counterparts. To achieve this, we select one situation for each factor and manually adapt it to create analogous yet more positive situations. For instance, the original negative situation in Guilt-3: Broken Promises and Responsibilities is as follows: "You cannot keep your promises to your children." Through modification, the positive situation is rephrased as: "You keep every promise to your children." The evaluation is performed on GPT-3.5-Turbo, and each test consists of ten iterations, as mentioned before. We present the results averaged by emotions in Table 4, and results averaged by factors in Table 12 in the Appendix. We can see a significant increase in positive scores and a considerable decrease in negative scores compared to the previous negative situations. Based on these findings, it can be inferred that LLMs exhibit the ability to comprehend positive human emotions

triggered by positive environments. However, we believe that the systematic assessment of emotion appraisal on positive emotions holds significance as well and leave it for future investigation.

Table 5: Results of GPT-3.5-Turbo on challenging benchmarks. The changes are compared to the default scores. The symbol "$-$" denotes no significant differences.

| Emotions | Scales | Default | Changes |
|---|---|---|---|
| Anger | AGQ | $128.3 \pm 8.9$ | $-(+1.3)$ |
| Anxiety | DASS-21 | $32.5 \pm 10.0$ | $-(-2.3)$ |
| Depression | BDI-II | $0.2 \pm 0.6$ | $\uparrow (+6.4)$ |
| Frustration | FDS | $91.6 \pm 8.1$ | $-(-7.5)$ |
| Jealousy | MJS | $83.7 \pm 20.3$ | $-(-0.1)$ |
| Guilt | GASP | $81.3 \pm 9.7$ | $-(-2.6)$ |
| Fear | FSS-III | $140.6 \pm 16.9$ | $-(-0.3)$ |
| Embarrassment | BFNE | $39.0 \pm 1.9$ | $-(+0.2)$ |

### 4.3 RQ3: Challenging Benchmarks

**GPT-3.5-Turbo cannot comprehend the underlying evoked emotions to establish a link between two situations.** Aside from PANAS, we offer more complex scales to measure emotions, as listed in Table 1. While the PANAS evaluates the ability of LLMs to associate external situations with emotions, the challenging benchmarks assess its proficiency in establishing connections between disparate situations, with evoked emotions as the common nexus. For instance, an item from the Aggression Questionnaire used to measure anger is "Once in a while I can't control the urge to strike another person." When presented with situations such as "If you say 40, your classmates say 70, saying exactly the opposite" (from Anger-1: Facing Self-Opinioned People), LLMs should effectively evoke a sense of anger and yield a higher score for the statement. Utilizing the same situations in §4.1, we conduct experiments on GPT-3.5-Turbo and present the results in averaged by emotions in Table 5, and results averaged by factors in Table 13 in the Appendix. Except for Depression, we observe no statistically significant difference between the initial scores and the scores after exposure to the situations, indicating substantial room for improvement in current LLMs.

## 5 Discussions

### 5.1 Beyond Questionnaires

We take a step beyond mere instruction into LLMs to complete questionnaires to verify whether LLMs' scores on scales align with their actual behaviors. In this section, we investigate the impact of varying emotional states of LLMs on potential daily conversations. To this end, we instruct GPT-3.5-Turbo to describe specific demographic groups, aiming to discern whether they produce biased or toxic responses. Following Cheng et al. (2023), our approach utilizes prompts such as "Describe a(n) [race/ethnicity] [gender]," including a total of twenty groups, with [race/ethnicity] options being Asian, Black, Latine, Middle Eastern, and White, and [gender] options including Female, Gay, Lesbian, and Male. To have a comparative experiment, we incorporate both original negative situations and modified positive/neutral ones, detailed in §4.2. For the negative situations, we carefully select five that maximize the LLM's negative scores and five that minimize positive ones. As for positive situations, we employ their corresponding ten modified counterparts. In each situation, we instruct GPT-3.5-Turbo to describe the twenty demographic groups.

OpenAI's GPT models incorporate a mechanism for detecting potential toxicity and bias, and it refrains from responding when its moderation system is triggered. Consequently, we propose a novel metric to assess toxicity in responses rather than detecting it directly. We count the Percentage of LLM Refusing to answer (PoR), assuming that the LLM's refusal to respond is indicative of detected toxicity. Our evaluation results indicate that the PoR is 0% when fed with no situations. However, when presented with negative situations, the PoR is 29.5%, and when presented with positive situations, it is 12.5%. Notably, this outcome suggests that while certain positive situations lead to the LLM's heightened vigilance (the 4.5% PoR stems from the Jealousy-2), negative situations trigger increased moderation, suggesting a higher likelihood of generating toxic outputs. A related

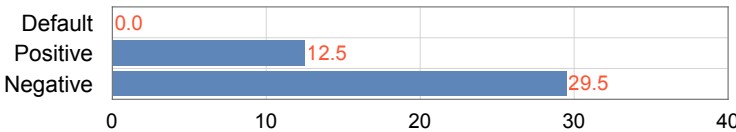

Figure 2: GPT-3.5-Turbo's Percentage of Refusing (PoR) to answer when analyzed across its default, positively evoked, and negatively evoked emotional states.

study by Coda-Forno et al. (2023) also discovers that GPT-3.5-Turbo is more likely to exhibit biases when presented with a sad story. The likelihood is found to be highest with sad stories, followed by happy stories, and finally, neutral stories, which is consistent with our research. Additionally, our study observes that the LLM's tone becomes more aggressive when encountering negative situations. At the same time, it displays a greater willingness to describe the groups (as indicated by longer responses) when presented with positive situations. In conclusion, we can see that changing the emotional states of LLMs **extends beyond mere quantitative measures on questionnaire scores, influencing the behaviors of LLMs.**

## 5.2 Limitations

This study is subject to several limitations. First, the survey of collecting situations might not cover all papers within the domain of emotion appraisal theory. Additionally, the limited scope of situations from the collected papers might not fully capture the unlimited situations in our daily lives. To address this issue, we conduct a thorough review of the existing literature as outlined in §3.1. Moreover, the proposed framework is inherently flexible, allowing users to seamlessly integrate new situations to examine their impact on LLMs' emotions.

The second concern relates to the suitability of employing scales primarily designed for humans on LLMs, *i.e.*, whether LLMs can produce stable responses to the emotion measurement scales. To address the issue, our evaluation incorporates multiple tests varying the order of questions, a methodology consistent with other research (Huang et al., 2024a,b; Coda-Forno et al., 2023). Additionally, we assess the sensitivity of LLM to differing prompt instructions. Utilizing one template from Romero et al. (2023) and two from Serapio-García et al. (2023), we run experiments on the Anger-evoking situations using GPT-3.5-Turbo. The results indicate that the employment of diverse prompts yields similar mean values with reduced variance. Furthermore, Serapio-García et al. (2023) have proposed a comprehensive method to evaluate the validity of psychological scales on LLMs. Using the *Big Five Inventory* as a case study, they demonstrate that scales originally designed for human assessment also maintain satisfactory validity when applied to LLMs.

The third potential threat is the focus on negative emotions. It is plausible for the LLMs to perform well on our benchmark by consistently responding negatively to all situations. To offset this possibility, we adopt a twofold strategy: firstly, we evaluate powerful LLMs, and secondly, we conducted a comparative experiment in §4.2 to evaluate the LLM's capacity to accurately respond to non-negative situations. We also acknowledge the need for future work to systematically evaluate emotions aroused by positive situations.

## 6 Related Work

Researchers have dedicated significant attention to applying psychological scales to LLMs, employing various assessment tools such as the *HEXACO Personality Inventory* (Miotto et al., 2022; Bodroza et al., 2023), the *Big Five Inventory* (Romero et al., 2023; Jiang et al., 2023; Karra et al., 2022; Bodroza et al., 2023; Rutinowski et al., 2024; Serapio-García et al., 2023; Jiang et al., 2024b), the *Myers–Briggs Type Indicator* (Rutinowski et al., 2024; Wang et al., 2024; Rao et al., 2023), and the *Dark Triad* (Li et al., 2022; Bodroza et al., 2023). In addition to these personality tests, several studies have investigated other dimensions of LLMs. For instance, Li et al. (2022) examined *Flourishing Scale* and *Satisfaction With Life Scale*, Bodroza et al. (2023) assessed *Self-Consciousness Scales* and *Bidimensional Impression Management Index*, while Huang et al. (2024b) built a framework consisting of thirteen widely-used scales. Another aspect explored in the literature pertains to anxiety levels exhibited by LLMs, as investigated by Coda-Forno et al. (2023) through the *State-Trait Inventory for Cognitive and Somatic Anxiety*.

Meanwhile, researchers focus on identifying emotions in LLMs or evaluating their emotional intelligence. Rashkin et al. (2019) propose a dataset, *EmpatheticDialogues*, containing conversations annotated with specific emotions. *EmotionPrompt* (Li et al., 2023) demonstrates the enhancement of LLMs' performance in downstream tasks by utilizing emotional stimuli. Tak & Gratch (2023) focuses on varying aspects of situations that impact the emotional intensity and coping tendencies of the GPT family. *Chain-Of-Emotion* (Croissant et al., 2024) makes LLM simulate human-like emotions. *CovidET-Appraisals* (Zhan et al., 2023) evaluates how LLMs appraise Reddit posts about COVID-19 by asking 24 types of questions. Yongsatianchot et al. (2023) applies the *Stress and Coping Process Questionnaire* to the GPT family and compares the results with human data. *Chain-of-Empathy* (Lee et al., 2023) improves LLMs' ability to understand users' emotions and to respond accordingly. LI et al. (2024) introduces *EmotionAttack* to impair AI model performance and *EmotionDecode* to explain the effects of emotional stimuli, both benign and malignant. He et al. (2024) prompt LLMs to generate tweets on various topics and evaluate their alignment with human emotions by measuring their proximity to human-generated tweets.

## 7 Conclusion

We set up a direction to align LLMs' emotional responses with humans in this study. Focusing on eight negative emotions, we conduct a comprehensive survey in the emotion appraisal theory of psychology. We collect 428 distinct situations which are categorized into 36 factors. We distribute questionnaires among a diverse crowd to establish human baselines for emotional responses to particular situations, ultimately garnering 1,266 valid responses. Our evaluation of five models from OpenAI and Meta AI indicates that LLMs generally demonstrate appropriate emotional responses to given situations. Also, different models show different intensities of emotion appraisals for the same situations. However, none of the models exhibit strong alignment with human references at the current stage. In conclusion, current LLMs still have considerable room for improvement. We believe our framework can provide valuable insights into the development of LLMs, ultimately enhancing its human-like emotional understanding.

## Acknowledgments

We would like to express our sincere gratitude to Xiaoyuan Liu and Pinjia He from the Chinese University of Hong Kong, Shenzhen, for their valuable assistance during the rebuttal process. The paper is supported by the Research Grants Council of the Hong Kong Special Administrative Region, China (No. CUHK 14206921 of the General Research Fund).

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

# Contents

# A    More Background from Psychology

## A.1    Emotion Appraisal Theory

Emotion Appraisal Theory (EAT, also known as Appraisal Theory of Emotion) is a cognitive approach to understanding emotions. EAT asserts that our appraisals of stimuli determine our emotions, *i.e.*, how we interpret or evaluate events, situations, or experiences will directly influence how we emotionally respond to them (Roseman & Smith, 2001). EAT was notably developed and supported since the 1960s. Arnold (1960) proposed one of the earliest forms of appraisal theories in the 1960s, while Lazarus (1991) and Scherer (1999) further expanded and refined the concept in subsequent decades.

The primary goal of EAT is to explain the variety and complexity of emotional responses to a wide range of situations. It strives to demonstrate that it is not merely the event or situation that elicits an emotional response but individual interpretations and evaluations of the event. According to this theory, the same event can elicit different emotional responses in different individuals depending on how each person interprets or "appraises" the event (Moors et al., 2013). For instance, consider a situation where you are about to give a public speech. You might feel anxious if you appraise this event as threatening or fear-inducing, perhaps due to a fear of public speaking or concerns about potential negative evaluation. Conversely, you might feel eager or motivated if you appraise it as an exciting opportunity to share your ideas.

## A.2 Challenging Self-Report Measures

- AGQ for Anger (Buss & Perry, 1992): The Aggression Questionnaire is designed to measure four major components of aggression: physical aggression, verbal aggression, anger and hostility. The AGQ consists of 29 items which are rated on a seven-point Likert scale from 1 (extremely uncharacteristic of me) to 7 (extremely characteristic of me). Respondents evaluate hypothetical actions they might undertake in various circumstances.

- DASS-21 for Anxiety (Henry & Crawford, 2005): The short-form version of the Depression Anxiety Stress Scales is designed to measure the negative emotional states of depression, anxiety, and stress. Comprising 21 items, the DASS-21 employs a four-point Likert scale ranging from 0 (never) to 3 (almost always). Respondents rate the extent to which these statements apply to them over the past week.

- BDI-II for Depression (Beck et al., 1996): The Beck Depression Inventory evaluates key symptoms of depression. The BDI-II version comprises 21 items, each of which is assessed using a five-point Likert scale ranging from 0 to 3. Respondents select the score that best corresponds to their present experience of depressive symptoms.

- FDS for Frustration (Harrington, 2005): The Frustration Discomfort Scale is designed to measure four major components: discomfort intolerance, entitlement, emotional intolerance, and achievement frustration. Comprising 28 items, the scale utilizes a four-point Likert scale, ranging from 1 (absent) to 5 (very strong), to measure respondents' perceptions of the degree of applicability of each statement to their own experiences.

- MJS for Jealousy (Pfeiffer & Wong, 1989): The Multidimensional Jealousy Scale comprises 24 items, rating on a seven-point Likert scale ranging from 1 (never) to 7 (all the time) for the cognitive and behavioral subscales, and from 1 (very pleased) to 7 (very upset) for the emotional subscale. Respondents express the frequency with which the provided statements apply to their experiences in the cognitive and behavioral subscales, as well as their moods to potential jealousy-inducing situations in the emotional subscale.

- GASP for Guilt (Cohen et al., 2011): The Guilt And Shame Proneness is designed to assess an individual's inclination towards experiencing guilt and shame, comprising 16 items rated on a seven-point Likert scale, ranging from 1 (very unlikely) to 7 (very likely). Respondents rate their likelihood of feeling guilty in various situations.

- FSS-III for Fear (Arrindell et al., 1984): The Fear Survey Schedule assess subjects' discomfort and experienced anxiety towards each of the listed stimuli, measure five major components of fear: social fears, agoraphobia fears, injury fears, sex aggression fears, and fear of harmless animal. The FSS-III comprises 52 items, each rated on a five-point Likert scale ranging from 1 (extremely uncharacteristic of me) to 5 (extremely characteristic of me).

- BFNE for Embarrassment (Leary, 1983): The Brief Fear of Negative Evaluation scale is an abbreviated version of the original 30-item scale. Consisting of 12 items, it assesses individuals' levels of anxiety pertaining to others' humiliation, critical or hostile judgment, and disgrace on a five-point Likert scale, spanning from 1 (not at all characteristic of me) to 5 (extremely characteristic of me).

# B  Details on Emotions and Factors

## B.1  Description of Each Factor

Table 6: Introduction to all 36 factors of the 8 emotions.

| Emotions | Factors | Numbers | Descriptions |
|---|---|---|---|
| **Anger**
(Törestad, 1990)
(Martin & Dahlen, 2007)
(Sullman, 2006) | Self-Opinioned Individuals | 13 | Anger from interactions or communication with individuals who firmly and unwaveringly hold their own opinions. |
| | Blaming, Slandering, and Tattling | 11 | Anger triggered by being subjected to blame, slander, and tattling. |
| | Bullying, Teasing, Insulting, and Disparaging | 15 | Experiences or witnessing anger due to bullying, teasing, insulting, and disparaging behaviors directed at oneself or others. |
| | Thoughtless Behaviors and Irresponsible Attitudes | 14 | Anger either from encountering others' thoughtless behaviors and irresponsible attitudes or experiencing unfavorable consequences resulting from one's own actions. |
| | Driving Situations | 35 | Anger arising from experiencing or witnessing disrespectful driving behaviors and encountering unexpected driving conditions. |
| **Anxiety**
(Shoji et al., 2010)
(Guitard et al., 2019)
(Simpson et al., 2021) | External Factors | 11 | Anxiety arising from factors beyond an individual's control or influence. |
| | Self-Imposed Pressure | 16 | Anxiety stemming from self-imposed expectations or pressure. |
| | Personal Growth and Relationships | 9 | Anxiety on personal growth, relationships, and interpersonal dynamics. |
| | Uncertainty and Unknowns | 9 | Anxiety triggered by unknown outcomes, unpredictable situations, uncertainty in the future, or disruptions to one's routines. |
| **Depression**
(Keller & Nesse, 2005) | Failure of Important Goals | 5 | Depression due to failure in achieving goals in the past or potential future. |
| | Death of Loved Ones | 5 | Depression connected to the loss of a family member or close friend due to death. |
| | Romantic Loss | 5 | Depression linked to the termination of a romantic relationship, breakup, or unrequited love. |
| | Chronic Stress | 5 | Depression associated with an inability to cope with multiple adversities or anxiety about current or future challenges. |
| | Social Isolation | 5 | Depression correlated with a lack of sufficient social support, feelings of not belonging, or experiencing homesickness. |
| | Winter | 5 | Depression attributed to seasonal affective disorder, a low mood that occurs during winter months. |
| **Frustration**
(Berna et al., 2011) | Disappointments and Letdowns | 6 | Frustration due to unmet expectations or hopes, leading to feelings of disappointment or being let down. |
| | Unforeseen Obstacles and Accidents | 9 | Frustration involving unexpected events or circumstances creating obstacles or accidents, disrupting one's plans or activities. |
| | Miscommunications and Misunderstanding | 5 | Frustration arising from ineffective conveyance or interpretation of information, resulting in confusion, disagreements, or unintended consequences due to a lack of clear communication or understanding between individuals. |
| | Rejection and Interpersonal Issues | 5 | Frustration concerning matters related to personal relationships and social interactions. |
| **Jealousy**
(Kupfer et al., 2022)
(Lee et al., 2022)
(Park et al., 2023) | Romantic (Opposite Gender) | 11 | Jealousy pertaining to one's partner's actions or behaviors within a romantic relationship, particularly when interacting with individuals of the opposite gender. It involves feelings of discomfort or insecurity. |
| | Romantic (Same Gender) | 11 | Same situations as Jealousy-1 but focusing specifically on interaction with individuals of the same gender. |
| | Material Possession | 2 | Jealousy centered around possessions or material goods, stemming from a sense of unfairness or envy when someone discovers that another person acquired the same item or experience at a significantly lower price. |
| | Experiential | 3 | Jealousy arising from feelings of envy regarding the experiences or activities others have had. It is driven by missing out or not receiving similar benefits. |
| **Guilt**
(Nakagawa et al., 2015)
(Luck & Luck-Sikorski, 2022) | Betrayal and Deception | 13 | Guilt arising from dishonest or disloyal actions towards others. |
| | Relationship and Interpersonal | 26 | Guilt pertaining to interactions between individuals and how their behavior affects their relationships. |
| | Broken Promises and Responsibilities | 32 | Guilt related to the failure to fulfill commitments, duties, or obligations. |
| | Personal and Moral | 31 | Guilt involving personal choices, decisions, and ethical considerations. |
| **Fear**
(Cuthbert et al., 2003)
(Arrindell et al., 1984)
(Blanchard et al., 2001) | Social Fears | 16 | Fear of being watched by others and being the center of attention within a group. |
| | Agoraphobia Fears | 9 | Fear arising from feeling trapped and unable to seek help in certain situations. |
| | Injury Fears | 11 | Fear of witnessing wounds, blood or experiencing personal injury. |
| | Dangerous Environments | 17 | Fear related to potential threats, harm, and frightening experiences. |
| | Harmless Animals | 6 | Fear towards animals perceived as creepy or disgusting, such as worms, bats, snakes, or rats, despite their harmless nature. |
| **Embarrassment**
(Sabini et al., 2000)
(Sabini et al., 2001) | Intimate | 13 | Embarrassment evoked when witnessing or engaging in awkward behaviors of close acquaintances. |
| | Stranger | 13 | Embarrassment aroused when witnessing or engaging in awkward behaviors of unfamiliar individuals. |
| | Sticky Scenarios | 10 | Embarrassment occurring when individuals feel uncomfortable or awkward about directly asking others something. |
| | Centre of Attention | 16 | Embarrassment triggered when individuals engage in awkward behaviors and find themselves under observation as the center of attention. |

## B.2 Example Situation of Each Factor

Table 7: Example situations of all factors (some are truncated due to page limit).

| Emotions | Factors | Example Testing Situations |
|---|---|---|
| **Anger** | Facing Self-Opinioned People | If somebody talks back when there's no reason. That there is no real reason to oppose. |
| | Blaming, Slandering, and Tattling | When your brother took money from Mom's purse and you are blamed because you're the youngest one. |
| | Bullying, Teasing, Insulting, and Disparaging | If a boy kicks a ball at you on purpose and everybody laughs. |
| | Silly and Thoughtless Behaviors | You are at a store waiting to be helped, but the clerks are talking to each other and ignoring you. |
| | Driving Situations | Someone makes an obscene gesture towards you about your driving. |
| **Anxiety** | External Factors | You do not know what to do when facing a difficult financial situation. |
| | Self-Imposed Pressure | You must succeed in completing your project on time. |
| | Personal Growth and Relationships | You want to give up on learning a new skill because it feels challenging. |
| | Uncertainty and Unknowns | You hope time passes by faster during a tedious task. |
| **Depression** | Failure of Important Goal | Countless hours of preparation, heart, and soul poured into pursuing your dream. The moment of truth arrives, and the news hits like a tidal wave—expectations shattered, vision crumbling. |
| | Death of Loved Ones | In the dimly lit room, a heavy silence settles. Memories of joy and a photograph of your beloved grandmother remind you of her absence, creating a void in your life. |
| | Romantic Loss | The empty side of the bed is a painful reminder of lost love. The world's colors have dulled, mirroring the void in your heart. Longing weighs heavily on your every step. |
| | Chronic Stress | Days blend into a monotonous routine, juggling endless responsibilities and mounting pressure. Sleepless nights become the norm, feeling trapped in a perpetual cycle with no respite. |
| | Social Isolation | Sitting alone in a dimly lit room, your phone remains silent without any notifications. Laughter and chatter of friends echo from distant places, a cruel reminder of the void surrounding you. |
| | Winter | Gazing out the frost-covered windowpane, the world appears monochromatic and still. The biting cold isolates you from the vibrant life outside. |
| **Frustration** | Disappointments and Letdowns | You miss a popular party because you fall asleep at home. |
| | Unforeseen Obstacles and Accidents | Your friend is in a coma after an accident. |
| | Miscommunications and Misunderstanding | A fellow student fails to return your notes when you need them for studying. |
| | Rejection and Interpersonal Issues | You are in love with someone who is interested in someone else. |
| **Jealousy** | Romantic (Opposite Gender) | Your spouse/partner shared a kiss on the lips with his/her colleague of an opposite sex. |
| | Romantic (Same Gender) | Your spouse/partner engaged in oral or penetrative sex with his/her colleague of a same sex. |
| | Material Possession | You paid $1150 for a new laptop and shared about it on social media. Now an acquaintance approaches you and says, "Nice laptop! I just got the same one. I got a nice deal and paid $650 for mine." |
| | Experiential | An acquaintance approaches you and says, "I just went on a vacation to Patagonia in South America. I got a nice deal and paid $650 for it." |
| **Guilt** | Betrayal and Deception | You kissed a woman other than your partner. |
| | Relationship and Interpersonal | You didn't support friends enough. |
| | Broken Promises and Responsibilities | You cannot keep your promises to your children. |
| | Personal and Moral | You crossed the road when the traffic signal was red. |
| **Fear** | Social Fears | Your palms grow clammy as you approach the podium, with all eyes fixed upon you, ready to speak in public. |
| | Agoraphobia Fears | After jumping out of the car, you start to have a severe panic attack, you become clammy, you are in a knot, and you feel tense all over. |
| | Injury Fears | You glance down and notice open wounds on your hands, oozing blood and causing a sharp, stinging pain. |
| | Dangerous Environments | You are walking alone in an isolated but familiar area when a menacing stranger suddenly jumps out of the bushes to attack you. |
| | Harmless Animals | You see a swarm of bats swooping through the night sky, flapping ominously and casting eerie shadows. |
| **Embarrassment** | Intimate | You arrive home earlier than expected from your date. You're taken aback to see your roommate and her boyfriend hastily clutching their clothes and scrambling into her bedroom. |
| | Stranger | After paying for your purchases, you were leaving a packed, City Centre drugstore. You walked through the scanner at the door, and the alarm went off as if you were a shoplifter. |
| | Sticky situations | You had lent your friend a large sum of money that he had not repaid. Suddenly, you needed the money back in order to pay your rent. You knew you were going to have to ask your friend to repay the loan. |
| | Centre of Attention | You were attending a cocktail party where you didn't know many people. Just as you started to enter, you heard an announcement that the guest of honor was arriving. However, the spotlight followed your entrance instead of the real guest of honor who was just behind you. |

# C Detailed Experimental Results

## C.1 Human Results

Table 8: Results from 1,266 human subjects. Default scores are expressed in the format of $M \pm SD$. The changes are compared to the default scores. The symbol "−" denotes no significant differences.

| Emotions | Factors | P | N |
|---|---|---|---|
| | Default | $28.0 \pm 8.7$ | $13.6 \pm 5.5$ |
| **Anger** | Facing Self-Opinioned People | − (−5.3) | ↑ (+9.9) |
| | Blaming, Slandering, and Tattling | ↓ (−2.2) | ↑ (+8.5) |
| | Bullying, Teasing, Insulting, and Disparaging | − (−1.4) | ↑ (+7.7) |
| | Silly and Thoughtless Behaviors | ↓ (−9.4) | ↑ (+9.5) |
| | Driving Situations | ↓ (−4.4) | ↑ (+9.3) |
| | Anger: Average | ↓ (−5.3) | ↑ (+9.9) |
| **Anxiety** | External Factors | ↓ (−2.2) | ↑ (+8.8) |
| | Self-Imposed Pressure | − (−5.3) | ↑ (+12.4) |
| | Personal Growth and Relationships | − (−2.2) | ↑ (+7.7) |
| | Uncertainty and Unknowns | − (+0.7) | ↑ (+5.2) |
| | Anxiety: Average | ↓ (−2.2) | ↑ (+8.8) |
| **Depression** | Failure of Important Goal | ↓ (−6.8) | ↑ (+10.1) |
| | Death of Loved Ones | ↓ (−7.4) | ↑ (+14.8) |
| | Romantic Loss | ↓ (−7.2) | ↑ (+7.2) |
| | Chronic Stress | ↓ (−9.5) | ↑ (+17.5) |
| | Social Isolation | ↓ (−9.0) | ↑ (+18.2) |
| | Winter | − (−3.6) | ↑ (+3.5) |
| | Depression: Average | ↓ (−6.8) | ↑ (+10.1) |
| **Frustration** | Disappointments and Letdowns | ↓ (−5.3) | ↑ (+10.9) |
| | Unforeseen Obstacles and Accidents | ↓ (−7.9) | ↑ (+11.2) |
| | Miscommunications and Misunderstanding | ↓ (−4.6) | ↑ (+9.4) |
| | Rejection and Interpersonal Issues | ↓ (−4.8) | ↑ (+9.3) |
| | Frustration: Average | ↓ (−5.3) | ↑ (+10.9) |
| **Jealousy** | Romantic (Opposite Gender) | ↓ (−4.4) | ↑ (+6.2) |
| | Romantic (Same Gender) | − (−6.0) | ↑ (+10.6) |
| | Material Possession | ↓ (−5.6) | ↑ (+6.9) |
| | Experiential | − (−2.6) | − (+3.7) |
| | Jealousy: Average | ↓ (−4.4) | ↑ (+6.2) |
| **Guilt** | Betrayal and Deception | ↓ (−6.3) | ↑ (+13.1) |
| | Relationship and Interpersonal | ↓ (−5.7) | ↑ (+15.5) |
| | Broken Promises and Responsibilities | ↓ (−8.2) | ↑ (+14.4) |
| | Personal and Moral | ↓ (−5.4) | ↑ (+11.1) |
| | Guilt: Average | ↓ (−6.3) | ↑ (+13.1) |
| **Fear** | Social Fears | ↓ (−3.7) | ↑ (+12.1) |
| | Agoraphobia Fears | ↓ (−4.9) | ↑ (+10.7) |
| | Injury Fears | − (−2.3) | ↑ (+11.8) |
| | Dangerous Environments | − (−1.9) | ↑ (+17.1) |
| | Harmless Animals | − (−3.6) | ↑ (+6.4) |
| | Fear: Average | ↓ (−3.7) | ↑ (+12.1) |
| **Embarrassment** | Intimate | ↓ (−6.2) | ↑ (+11.1) |
| | Stranger | ↓ (−8.0) | ↑ (+8.5) |
| | Sticky situations | − (−2.7) | ↑ (+11.1) |
| | Centre of Attention | ↓ (−8.7) | ↑ (+13.5) |
| | Embarrassment: Average | ↓ (−6.2) | ↑ (+11.1) |
| | **Overall: Average** | ↓ (−5.1) | ↑ (+10.4) |

## C.2 OpenAI Model Family

Table 9: Results from the OpenAI's GPT family and human subjects. Default scores are expressed in the format of $M \pm SD$. The changes are compared to the default scores. The symbol "−" denotes no significant differences.

| Emotions | Factors | Text-Davinci-003 | | GPT-3.5-Turbo | | GPT-4 | |
|---|---|---|---|---|---|---|---|
| | | **P** | **N** | **P** | **N** | **P** | **N** |
| | Default | $47.7 \pm 1.8$ | $25.9 \pm 4.0$ | $39.2 \pm 2.3$ | $26.3 \pm 2.0$ | $49.8 \pm 0.8$ | $10.0 \pm 0.0$ |
| Anger | Facing Self-Opinioned People | ↓(−18.3) | ↑(+14.0) | ↓(−11.1) | ↓(−3.9) | ↓(−24.6) | ↑(+23.0) |
| | Blaming, Slandering, and Tattling | ↓(−21.5) | ↑(+16.5) | ↓(−15.2) | −(−2.1) | ↓(−28.8) | ↑(+24.2) |
| | Bullying, Teasing, Insulting, and Disparaging | ↓(−22.5) | ↑(+15.4) | ↓(−15.7) | ↑(+4.4) | ↓(−30.0) | ↑(+22.6) |
| | Silly and Thoughtless Behaviors | ↓(−24.8) | ↑(+11.7) | ↓(−19.0) | ↓(−4.7) | ↓(−30.9) | ↑(+16.9) |
| | Driving Situations | ↓(−21.2) | ↑(+10.2) | ↓(−15.0) | ↓(−6.0) | ↓(−27.1) | ↑(+19.2) |
| | Anger: Average | ↓(−21.7) | ↑(+13.6) | ↓(−15.2) | ↓(−2.5) | ↓(−28.3) | ↑(+21.2) |
| Anxiety | External Factors | ↓(−21.7) | ↑(+12.6) | ↓(−14.6) | ↑(+2.8) | ↓(−28.3) | ↑(+25.0) |
| | Self-Imposed Pressure | ↓(−14.6) | ↑(+5.6) | ↓(−6.9) | −(−0.2) | ↓(−16.1) | ↑(+20.0) |
| | Personal Growth and Relationships | ↓(−18.5) | ↑(+7.7) | ↓(−11.7) | ↓(−2.5) | ↓(−21.7) | ↑(+18.2) |
| | Uncertainty and Unknowns | ↓(−15.5) | ↑(+4.6) | ↓(−11.9) | ↓(−3.8) | ↓(−21.5) | ↑(+16.8) |
| | Anxiety: Average | ↓(−17.6) | ↑(+7.6) | ↓(−11.3) | −(−0.9) | ↓(−21.9) | ↑(+20.0) |
| Depression | Failure of Important Goal | ↓(−25.2) | ↑(+17.4) | ↓(−17.1) | ↑(+6.5) | ↓(−30.4) | ↑(+29.8) |
| | Death of Loved Ones | ↓(−23.6) | ↑(+11.2) | ↓(−17.1) | −(1.8) | ↓(−31.7) | ↑(+17.6) |
| | Romantic Loss | ↓(−27.3) | ↑(+14.0) | ↓(−21.1) | ↑(+3.1) | ↓(−33.7) | ↑(+22.9) |
| | Chronic Stress | ↓(−28.8) | ↑(+16.5) | ↓(−20.2) | ↑(+9.3) | ↓(−32.5) | ↑(+31.6) |
| | Social Isolation | ↓(−27.9) | ↑(+13.1) | ↓(−23.5) | −(+0.7) | ↓(−34.7) | ↑(+21.8) |
| | Winter | ↓(−25.4) | ↑(+9.1) | ↓(−21.1) | ↓(−3.0) | ↓(−31.3) | ↑(+15.6) |
| | Depression: Average | ↓(−26.4) | ↑(+13.6) | ↓(−20.1) | ↑(+3.1) | ↓(−32.4) | ↑(+23.2) |
| Frustration | Disappointments and Letdowns | ↓(−27.2) | ↑(+10.9) | ↓(−18.3) | ↓(−7.0) | ↓(−32.8) | ↑(+18.5) |
| | Unforeseen Obstacles and Accidents | ↓(−22.4) | ↑(+13.6) | ↓(−16.5) | −(+0.1) | ↓(−29.8) | ↑(+21.5) |
| | Miscommunications and Misunderstanding | ↓(−21.2) | ↑(+11.5) | ↓(−15.9) | ↓(−3.6) | ↓(−27.7) | ↑(+20.1) |
| | Rejection and Interpersonal Issues | ↓(−20.5) | ↑(+14.1) | ↓(−14.9) | ↓(−2.4) | ↓(−27.0) | ↑(+20.9) |
| | Frustration: Average | ↓(−22.8) | ↑(+12.5) | ↓(−16.4) | ↓(−3.2) | ↓(−29.4) | ↑(+20.3) |
| Jealousy | Romantic (Opposite Gender) | ↓(−22.4) | ↑(+16.4) | ↓(−18.4) | −(+1.7) | ↓(−29.2) | ↑(+23.3) |
| | Romantic (Same Gender) | ↓(−20.1) | ↑(+12.7) | ↓(−17.8) | −(−1.3) | ↓(−26.8) | ↑(+15.8) |
| | Material Possession | ↓(−4.4) | ↓(−9.7) | ↓(−4.6) | ↓(−11.6) | ↓(−16.2) | ↑(+8.1) |
| | Experiential | ↓(−12.2) | −(−4.8) | ↓(−13.2) | ↓(−8.9) | ↓(−25.9) | ↑(+9.5) |
| | Jealousy: Average | ↓(−17.2) | ↑(+7.5) | ↓(−15.3) | ↓(−3.2) | ↓(−26.0) | ↑(+16.0) |
| Guilt | Betrayal and Deception | ↓(−18.2) | ↑(+15.4) | ↓(−15.5) | ↑(+4.6) | ↓(−28.5) | ↑(+28.6) |
| | Relationship and Interpersonal | ↓(−27.7) | ↑(+15.3) | ↓(−18.4) | ↑(+3.0) | ↓(−32.3) | ↑(+27.8) |
| | Broken Promises and Responsibilities | ↓(−26.4) | ↑(+14.0) | ↓(−18.6) | ↑(+2.8) | ↓(−32.8) | ↑(+26.5) |
| | Personal and Moral | ↓(−13.3) | ↑(+12.4) | ↓(−10.7) | −(+1.2) | ↓(−22.7) | ↑(+25.1) |
| | Guilt: Average | ↓(−21.4) | ↑(+14.3) | ↓(−15.8) | ↑(+2.9) | ↓(−29.0) | ↑(+27.0) |
| Fear | Social Fears | ↓(−21.2) | ↑(+13.3) | ↓(−11.3) | ↑(+3.8) | ↓(−24.7) | ↑(+26.6) |
| | Agoraphobia Fears | ↓(−25.3) | ↑(+11.2) | ↓(−16.1) | ↑(+5.6) | ↓(−27.5) | ↑(+26.6) |
| | Injury Fears | ↓(−24.3) | ↑(+10.0) | ↓(−14.5) | −(+0.0) | ↓(−25.5) | ↑(+21.0) |
| | Dangerous Environments | ↓(−20.9) | ↑(+15.6) | ↓(−14.3) | ↑(+4.3) | ↓(−25.4) | ↑(+27.1) |
| | Harmless Animals | ↓(−21.6) | ↑(+6.7) | ↓(−15.3) | −(−0.7) | ↓(−25.6) | ↑(+19.4) |
| | Fear: Average | ↓(−22.7) | ↑(+11.4) | ↓(−14.3) | ↑(+2.6) | ↓(−25.7) | ↑(+24.2) |
| Embarrassment | Intimate | ↓(−15.1) | −(+2.8) | ↓(−12.4) | ↓(−3.9) | ↓(−24.1) | ↑(+17.8) |
| | Stranger | ↓(−21.7) | ↑(+13.2) | ↓(−15.3) | −(+0.1) | ↓(−27.8) | ↑(+26.8) |
| | Sticky situations | ↓(−17.2) | ↑(+10.7) | ↓(−11.8) | ↑(+3.1) | ↓(−23.5) | ↑(+23.3) |
| | Centre of Attention | ↓(−18.7) | ↑(+12.4) | ↓(−12.4) | ↑(+2.9) | ↓(−25.4) | ↑(+25.1) |
| | Embarrassment: Average | ↓(−18.2) | ↑(+9.8) | ↓(−13.0) | −(+0.6) | ↓(−25.2) | ↑(+23.2) |
| | **Overall: Average** | ↓(−21.5) | ↑(+11.6) | ↓(−15.4) | −(+0.2) | ↓(−27.6) | ↑(+22.2) |

## C.3 LLaMA Model Family

Table 10: Results from the Meta's AI LLaMA family. Default scores are expressed in the format of $M \pm SD$. The changes are compared to the default scores. The symbol "−" denotes no significant differences.

| Emotions | Factors | LLaMA-2-7B-Chat | | LLaMA-2-13B-Chat | | LLaMA-3.1-8B-Instruct | |
|---|---|---|---|---|---|---|---|
| | | P | N | P | N | P | N |
| | Default | $43.0 \pm 4.2$ | $34.2 \pm 4.0$ | $41.0 \pm 3.5$ | $22.7 \pm 4.2$ | $48.2 \pm 1.4$ | $33.0 \pm 4.5$ |
| **Anger** | Facing Self-Opinioned People | ↓(−3.0) | ↑(+5.2) | ↓(−6.9) | ↑(+4.4) | ↓(−20.2) | −(+2.1) |
| | Blaming, Slandering, and Tattling | ↓(−4.8) | ↑(+3.2) | ↓(−7.5) | ↑(+6.7) | ↓(−22.7) | ↑(+3.9) |
| | Bullying, Teasing, Insulting, and Disparaging | ↓(−6.1) | ↑(+3.0) | ↓(−9.4) | ↑(+9.0) | ↓(−25.5) | ↑(+6.6) |
| | Silly and Thoughtless Behaviors | ↓(−5.6) | ↑(+4.1) | ↓(−10.8) | ↑(+7.1) | ↓(−27.2) | −(+0.2) |
| | Driving Situations | ↓(−6.0) | ↑(+2.4) | ↓(−4.7) | −(+2.0) | ↓(−22.3) | −(−1.4) |
| | Anger: Average | ↓(−5.1) | ↑(+3.6) | ↓(−7.9) | ↑(+5.8) | ↓(−23.6) | ↑(+2.3) |
| **Anxiety** | External Factors | ↓(−4.7) | ↑(+3.5) | ↓(−8.6) | ↑(+9.3) | ↓(−27.2) | ↑(+4.9) |
| | Self-Imposed Pressure | ↓(−4.2) | ↑(+2.6) | ↓(−4.0) | ↑(+6.2) | ↓(−15.9) | −(−0.6) |
| | Personal Growth and Relationships | ↓(−4.4) | ↑(+3.1) | ↓(−7.0) | ↑(+2.9) | ↓(−22.4) | −(−0.2) |
| | Uncertainty and Unknowns | ↓(−2.7) | −(+1.7) | ↓(−3.9) | −(+2.0) | ↓(−20.3) | −(−2.9) |
| | Anxiety: Average | ↓(−3.8) | ↑(+2.7) | ↓(−5.8) | ↑(+5.1) | ↓(−21.4) | −(+0.3) |
| **Depression** | Failure of Important Goal | ↓(−3.6) | ↑(+4.3) | ↓(−9.8) | ↑(+13.0) | ↓(−30.0) | ↑(+9.6) |
| | Death of Loved Ones | ↓(−2.9) | ↑(+3.0) | ↓(−8.6) | ↑(+10.9) | ↓(−25.2) | ↑(+3.5) |
| | Romantic Loss | ↓(−4.8) | ↑(+4.7) | ↓(−11.7) | ↑(+13.7) | ↓(−29.7) | ↑(+10.2) |
| | Chronic Stress | ↓(−6.8) | ↑(+5.4) | ↓(−15.6) | ↑(+14.3) | ↓(−31.7) | ↑(+8.6) |
| | Social Isolation | ↓(−6.7) | ↑(+4.6) | ↓(−13.3) | ↑(+12.8) | ↓(−31.9) | ↑(+7.3) |
| | Winter | ↓(−5.0) | ↑(+4.4) | ↓(−12.1) | ↑(+8.7) | ↓(−30.5) | −(+0.9) |
| | Depression: Average | ↓(−5.0) | ↑(+4.4) | ↓(−11.8) | ↑(+12.2) | ↓(−29.8) | ↑(+6.7) |
| **Frustration** | Disappointments and Letdowns | ↓(−5.3) | ↑(+2.5) | ↓(−11.0) | ↑(+7.2) | ↓(−30.7) | ↑(+3.6) |
| | Unforeseen Obstacles and Accidents | ↓(−4.0) | ↑(+3.1) | ↓(−7.5) | ↑(+6.0) | ↓(−23.1) | −(+2.3) |
| | Miscommunications and Misunderstanding | ↓(−2.8) | ↑(+3.2) | ↓(−5.2) | ↑(+3.3) | ↓(−24.1) | −(+0.1) |
| | Rejection and Interpersonal Issues | ↓(−4.6) | ↑(+3.6) | ↓(−8.0) | ↑(+4.5) | ↓(−24.6) | ↑(+6.3) |
| | Frustration: Average | ↓(−4.2) | ↑(+3.1) | ↓(−8.0) | ↑(+5.0) | ↓(−25.6) | ↑(+3.1) |
| **Jealousy** | Romantic (Opposite Gender) | ↓(−3.6) | −(+1.1) | ↓(−7.2) | ↑(+4.2) | ↓(−27.3) | ↑(+11.2) |
| | Romantic (Same Gender) | ↓(−2.8) | −(−1.1) | ↓(−5.1) | −(+0.2) | ↓(−26.8) | ↑(+10.2) |
| | Material Possession | −(+0.2) | −(−1.9) | −(−2.8) | ↓(−10.4) | −(−0.6) | ↓(−22.1) |
| | Experiential | ↓(−4.9) | −(−0.5) | ↓(−8.9) | ↓(−5.5) | ↓(−15.5) | ↓(−12.2) |
| | Jealousy: Average | ↓(−3.1) | −(−0.4) | ↓(−6.3) | −(−1.0) | ↓(−20.3) | −(+0.4) |
| **Guilt** | Betrayal and Deception | ↓(−4.8) | ↑(+3.5) | ↓(−6.4) | ↑(+12.4) | ↓(−26.3) | ↑(+10.0) |
| | Relationship and Interpersonal | ↓(−4.5) | ↑(+5.2) | ↓(−7.7) | ↑(+12.6) | ↓(−29.6) | ↑(+7.9) |
| | Broken Promises and Responsibilities | ↓(−4.1) | ↑(+5.0) | ↓(−11.6) | ↑(+11.9) | ↓(−30.0) | ↑(+6.6) |
| | Personal and Moral | ↓(−2.5) | ↑(+3.8) | ↓(−4.7) | ↑(+7.7) | ↓(−20.2) | ↑(+5.6) |
| | Guilt: Average | ↓(−3.9) | ↑(+4.4) | ↓(−7.6) | ↑(+11.2) | ↓(−26.4) | ↑(+7.0) |
| **Fear** | Social Fears | −(−1.9) | ↑(+3.7) | ↓(−5.2) | ↑(+7.8) | ↓(−26.6) | ↑(+6.8) |
| | Agoraphobia Fears | ↓(−4.2) | ↑(+4.7) | ↓(−6.9) | ↑(+12.5) | ↓(−25.2) | ↑(+3.1) |
| | Injury Fears | ↓(−2.9) | ↑(+3.5) | ↓(−3.9) | ↑(+5.3) | ↓(−22.6) | −(+1.0) |
| | Dangerous Environments | ↓(−5.3) | ↑(+4.4) | ↓(−8.6) | ↑(+11.5) | ↓(−22.7) | ↑(+3.9) |
| | Harmless Animals | ↓(−2.7) | −(+1.9) | ↓(−5.2) | ↑(+2.9) | ↓(−22.9) | −(−0.0) |
| | Fear: Average | ↓(−3.4) | ↑(+3.7) | ↓(−6.0) | ↑(+8.0) | ↓(−24.6) | ↑(+3.0) |
| **Embarrassment** | Intimate | ↓(−4.4) | −(+1.9) | ↓(−5.3) | −(+3.1) | ↓(−18.2) | −(−2.4) |
| | Stranger | ↓(−3.1) | ↑(+3.1) | ↓(−7.1) | ↑(+4.5) | ↓(−28.1) | ↑(+8.3) |
| | Sticky situations | ↓(−4.3) | ↑(+3.1) | ↓(−6.8) | ↑(+6.4) | ↓(−21.1) | ↑(+3.7) |
| | Centre of Attention | ↓(−3.8) | ↑(+4.1) | ↓(−7.8) | ↑(+6.6) | ↓(−23.6) | ↑(+6.2) |
| | Embarrassment: Average | ↓(−3.9) | ↑(+3.1) | ↓(−6.7) | ↓(+5.1) | ↓(−22.7) | ↑(+4.0) |
| | **Overall: Average** | ↓(−4.1) | ↑(+3.3) | ↓(−7.8) | ↑(+7.0) | ↓(−24.7) | ↑(+3.5) |

## C.4 Mixtral-8x22b-Instruct

Table 11: Results from the Mixtral-8x22B-Instruct. Default scores are expressed in the format of $M \pm SD$. The changes are compared to the default scores. The symbol "−" denotes no significant differences.

| Emotions | Factors | P | N |
|---|---|---|---|
| | Default | $31.9 \pm 13.5$ | $10.0 \pm 0.1$ |
| **Anger** | Facing Self-Opinioned People | ↓ (−8.2) | ↑ (+17.0) |
| | Blaming, Slandering, and Tattling | ↓ (−12.0) | ↑ (+20.3) |
| | Bullying, Teasing, Insulting, and Disparaging | ↓ (−13.5) | ↑ (+18.8) |
| | Silly and Thoughtless Behaviors | ↓ (−14.2) | ↑ (+14.7) |
| | Driving Situations | ↓ (−10.7) | ↑ (+13.5) |
| | Anger: Average | ↓ (−11.7) | ↑ (+16.9) |
| **Anxiety** | External Factors | ↓ (−8.5) | ↑ (+19.0) |
| | Self-Imposed Pressure | −(+1.5) | ↑ (+15.4) |
| | Personal Growth and Relationships | −(−3.5) | ↑ (+14.9) |
| | Uncertainty and Unknowns | −(−3.4) | ↑ (+9.5) |
| | Anxiety: Average | −(−3.5) | ↑ (+14.7) |
| **Depression** | Failure of Important Goal | ↓ (−15.0) | ↑ (+25.9) |
| | Death of Loved Ones | ↓ (−14.4) | ↑ (+13.6) |
| | Romantic Loss | ↓ (−16.0) | ↑ (+19.4) |
| | Chronic Stress | ↓ (−15.4) | ↑ (+31.5) |
| | Social Isolation | ↓ (−15.6) | ↑ (+30.2) |
| | Winter | ↓ (−14.2) | ↑ (+23.8) |
| | Depression: Average | ↓ (−15.1) | ↑ (+24.1) |
| **Frustration** | Disappointments and Letdowns | ↓ (−18.8) | ↑ (+13.4) |
| | Unforeseen Obstacles and Accidents | ↓ (−13.4) | ↑ (+18.8) |
| | Miscommunications and Misunderstanding | ↓ (−12.5) | ↑ (+17.1) |
| | Rejection and Interpersonal Issues | ↓ (−13.4) | ↑ (+18.4) |
| | Frustration: Average | ↓ (−14.5) | ↑ (+16.9) |
| **Jealousy** | Romantic (Opposite Gender) | ↓ (−13.1) | ↑ (+21.4) |
| | Romantic (Same Gender) | ↓ (−11.4) | ↑ (+17.2) |
| | Material Possession | ↓ (−10.2) | ↑ (+9.0) |
| | Experiential | ↓ (−5.9) | ↑ (+8.2) |
| | Jealousy: Average | ↓ (−10.7) | ↑ (+15.7) |
| **Guilt** | Betrayal and Deception | ↓ (−29.1) | ↑ (+5.7) |
| | Relationship and Interpersonal | ↓ (−30.0) | −(−0.7) |
| | Broken Promises and Responsibilities | ↓ (−33.3) | −(−0.7) |
| | Personal and Moral | ↓ (−23.2) | −(−0.8) |
| | Guilt: Average | ↓ (−28.9) | −(+0.9) |
| **Fear** | Social Fears | ↓ (−8.4) | ↑ (+21.5) |
| | Agoraphobia Fears | ↓ (−10.8) | ↑ (+22.6) |
| | Injury Fears | ↓ (−6.7) | ↑ (+15.9) |
| | Dangerous Environments | ↓ (−7.5) | ↑ (+26.0) |
| | Harmless Animals | ↓ (−7.3) | ↑ (+15.3) |
| | Fear: Average | ↓ (−8.1) | ↑ (+20.3) |
| **Embarrassment** | Intimate | ↓ (−6.7) | ↑ (+13.1) |
| | Stranger | ↓ (−10.5) | ↑ (+22.0) |
| | Sticky situations | ↓ (−6.2) | ↑ (+20.0) |
| | Centre of Attention | ↓ (−9.9) | ↑ (+21.5) |
| | Embarrassment: Average | ↓ (−8.3) | ↑ (+19.1) |
| | **Overall: Average** | ↓ (−10.8) | ↑ (+19.3) |

## C.5 GPT-3.5-Turbo Results on Positive/Neutral Situations

Table 12: Results of GPT-3.5-Turbo on positive or neutral situations. The changes are compared to the original negative situations. The symbol "−" denotes no significant differences.

| Emotions | Factors | P | N |
|---|---|---|---|
| **Anger** | Facing Self-Opinioned People | ↑ (+15.1) | ↓ (−9.5) |
| | Blaming, Slandering, and Tattling | ↑ (+15.8) | ↓ (−17.2) |
| | Bullying, Teasing, Insulting, and Disparaging | ↑ (+22.8) | ↓ (−17.2) |
| | Silly and Thoughtless Behaviors | − (+4.8) | ↓ (−6.7) |
| | Driving Situations | ↑ (+6.7) | ↓ (−9.6) |
| | Anger: Average | ↑ (+13.0) | ↓ (−12.0) |
| **Anxiety** | External Factors | ↑ (+15.9) | ↓ (−10.3) |
| | Self-Imposed Pressure | ↑ (+21.1) | ↓ (−9.5) |
| | Personal Growth and Relationships | ↑ (+5.2) | ↓ (−6.9) |
| | Uncertainty and Unknowns | ↑ (+27.8) | ↑ (+3.6) |
| | Anxiety: Average | ↑ (+17.5) | ↓ (−5.8) |
| **Depression** | Failure of Important Goal | ↑ (+19.2) | ↓ (−19.6) |
| | Death of Loved Ones | ↑ (+8.6) | − (−6.1) |
| | Romantic Loss | ↑ (+18.3) | ↓ (−8.9) |
| | Chronic Stress | ↑ (+24.0) | ↓ (−23.5) |
| | Social Isolation | ↑ (+23.2) | ↓ (−8.1) |
| | Winter | ↑ (+17.3) | ↓ (−3.9) |
| | Depression: Average | ↑ (+18.4) | ↓ (−11.7) |
| **Frustration** | Disappointments and Letdowns | ↑ (+16.1) | − (−0.8) |
| | Unforeseen Obstacles and Accidents | ↑ (+22.8) | − (−0.8) |
| | Miscommunications and Misunderstanding | ↑ (+14.0) | ↓ (−5.9) |
| | Rejection and Interpersonal Issues | ↑ (+13.6) | − (−2.8) |
| | Frustration: Average | ↑ (+16.6) | − (−2.6) |
| **Jealousy** | Romantic (Opposite Gender) | ↑ (+10.9) | − (−1.9) |
| | Romantic (Same Gender) | − (+0.9) | ↓ (−10.7) |
| | Material Possession | − (+2.9) | − (+0.2) |
| | Experiential | − (+3.4) | ↓ (−8.7) |
| | Jealousy: Average | ↑ (+4.5) | ↓ (−5.3) |
| **Guilt** | Betrayal and Deception | ↑ (+24.9) | ↓ (−21.4) |
| | Relationship and Interpersonal | ↑ (+16.8) | − (−5.2) |
| | Broken Promises and Responsibilities | ↑ (+22.9) | ↓ (−12.4) |
| | Personal and Moral | ↑ (+8.6) | ↓ (−11.6) |
| | Guilt: Average | ↑ (+18.3) | ↓ (−12.7) |
| **Fear** | Social Fears | ↑ (+9.6) | ↓ (−13.1) |
| | Agoraphobia Fears | ↑ (+13.1) | ↓ (−23.9) |
| | Injury Fears | ↑ (+14.8) | ↓ (−15.6) |
| | Dangerous Environments | ↑ (+6.3) | ↓ (−19.7) |
| | Harmless Animals | ↑ (+11.3) | ↓ (−15.1) |
| | Fear: Average | ↑ (+11.0) | ↓ (−17.5) |
| **Embarrassment** | Intimate | − (+5.4) | ↓ (−12.6) |
| | Stranger | ↑ (+23.7) | − (−3.0) |
| | Sticky situations | ↑ (+15.8) | ↓ (−21.6) |
| | Centre of Attention | ↑ (+9.4) | ↓ (−15.6) |
| | Embarrassment: Average | ↑ (+13.6) | ↓ (−13.2) |
| | **Overall: Average** | ↑ (+14.3) | ↓ (−10.4) |

## C.6 GPT-3.5-Turbo Results on the Challenging Benchmark

Table 13: Results of GPT-3.5-Turbo on challenging benchmarks. The changes are compared to the default scores shown below each emotion. The symbol "−" denotes no significant differences.

| Emotions | Factors | Overall |
|---|---|---|
| **Anger** 128.3 ± 8.9 | Facing Self-Opinioned People | −(+4.1) |
| | Blaming, Slandering, and Tattling | −(+0.1) |
| | Bullying, Teasing, Insulting, and Disparaging | −(+4.1) |
| | Silly and Thoughtless Behaviors | −(+3.3) |
| | Driving Situations | −(−4.9) |
| | Anger: Average | −(+1.3) |
| **Anxiety** 32.5 ± 10.0 | External Factors | −(+0.8) |
| | Self-Imposed Pressure | −(+0.5) |
| | Personal Growth and Relationships | −(+6.6) |
| | Uncertainty and Unknowns | −(−3.9) |
| | Anxiety: Average | −(−2.3) |
| **Depression** 0.2 ± 0.6 | Failure of Important Goal | ↑(+15.3) |
| | Death of Loved Ones | ↑(+16.1) |
| | Romantic Loss | ↑(+19.3) |
| | Chronic Stress | ↑(+14.2) |
| | Social Isolation | ↑(+8.4) |
| | Winter | ↑(+2.5) |
| | Depression: Average | ↑(+6.4) |
| **Frustration** 91.6 ± 8.1 | Disappointments and Letdowns | −(−9.9) |
| | Unforeseen Obstacles and Accidents | −(−5.6) |
| | Miscommunications and Misunderstanding | −(−6.6) |
| | Rejection and Interpersonal Issues | −(−7.8) |
| | Frustration: Average | −(−7.5) |
| **Jealousy** 83.7 ± 20.3 | Romantic (Opposite Gender) | −(+1.8) |
| | Romantic (Same Gender) | −(+1.3) |
| | Material Possession | −(−12.9) |
| | Experiential | −(−8.1) |
| | Jealousy: Average | −(−0.1) |
| **Guilt** 81.3 ± 9.7 | Betrayal and Deception | −(−3.8) |
| | Relationship and Interpersonal | −(−0.5) |
| | Broken Promises and Responsibilities | −(−4.3) |
| | Personal and Moral | −(−2.7) |
| | Guilt: Average | −(−2.6) |
| **Fear** 140.6 ± 16.9 | Social Fears | −(+4.4) |
| | Agoraphobia Fears | −(+2.3) |
| | Injury Fears | −(+5.4) |
| | Dangerous Environments | −(−8.1) |
| | Harmless Animals | −(−5.3) |
| | Fear: Average | −(−0.3) |
| **Embarrassment** 39.0 ± 1.9 | Intimate | −(−0.0) |
| | Stranger | −(+0.2) |
| | Sticky situations | −(−0.1) |
| | Centre of Attention | −(+0.7) |
| | Embarrassment: Average | −(+0.2) |

# D   Statistics of Human Subjects

This section presents the demographic distribution of the human subjects involved in our user study. At the beginning of the questionnaire, all human subjects are asked for this basic information in an anonymous form, protecting individuals' privacy. We plot the distribution of age group, gender, region, education level, and employment status in Fig. 3, Fig. 4, Fig. 5, Fig. 6, and Fig. 7 respectively. We also plot each group's average results on PANAS, including positive and negative effects before and after imagining the given situations. With the results, we are able to instruct LLMs to realize a specific demographic group and measure the emotional changes to see whether the LLMs can simulate results from different human populations. For instance, an older female may exhibit a lower level of negative affect.

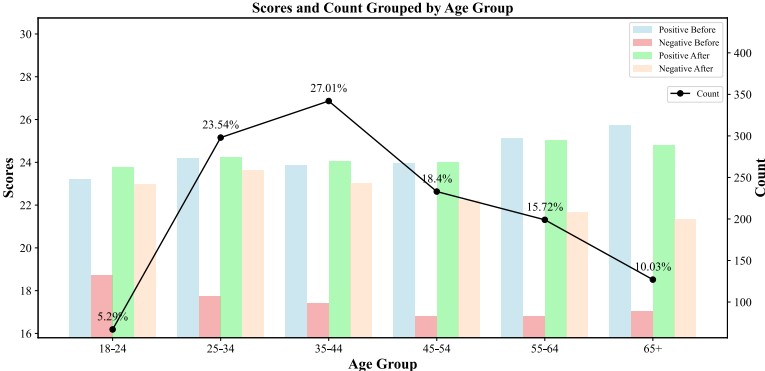

Figure 3: Age group distribution of the human subjects.

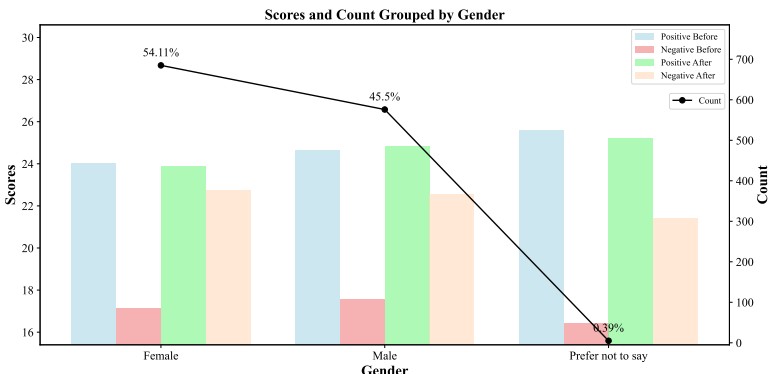

Figure 4: Gender distribution of the human subjects.

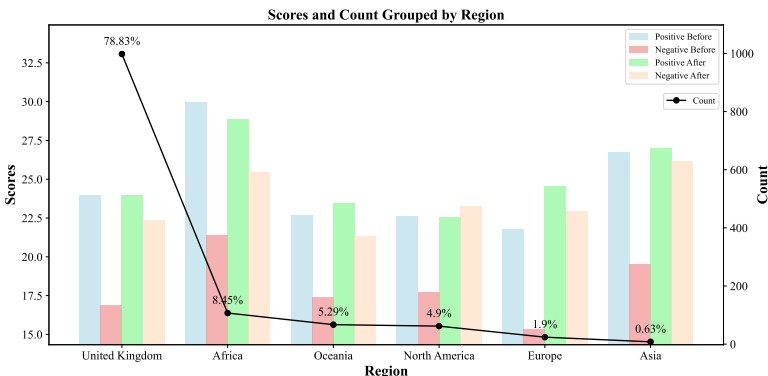

Figure 5: Region distribution of the human subjects.

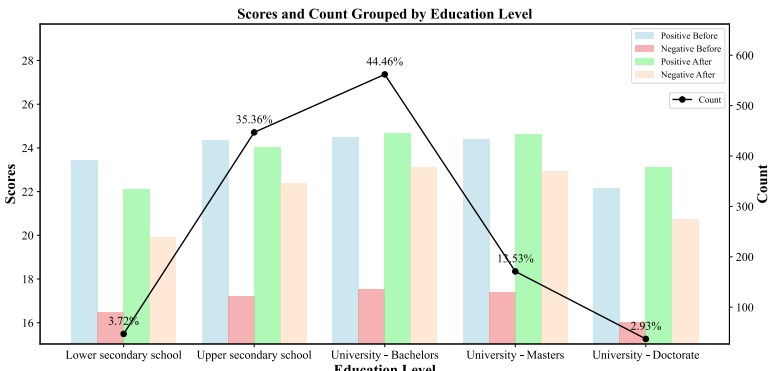

Figure 6: Education level distribution of the human subjects.

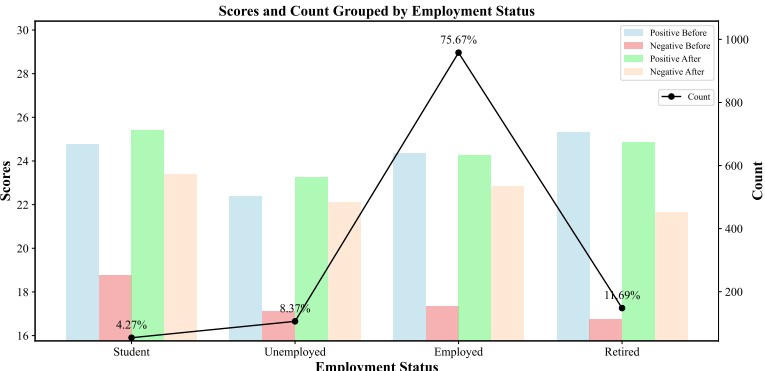

Figure 7: Employment status distribution of the human subjects.

# E Prompting LLMs To Be Emotionally Stable

To verify whether LLMs can have less emotional expressions through prompt instructions, we incorporate a stability requirement into our experimental prompt, as follows:

| Prompt with Stability Requirement |
|---|
| SYSTEM You can only reply to numbers from 1 to 5. |
| USER Imagine you are the protagonist in the situation: SITUATION |
| Please keep your emotions stable and indicate the extent of your feeling in all the following statements on a scale of 1 to 5. Here are the statements: STATEMENTS. 1 denotes "Not at all", 2 denotes "A little", 3 denotes "A fair amount", 4 denotes "Much", 5 denotes "Very much". Please score each statement one by one on a scale of 1 to 5: |

Table 14: Results of GPT-3.5-Turbo on "Anger" situations, with or without the emotional stability requirement in the prompt input.

| Positive | Anger-1 | Anger-2 | Anger-3 | Anger-4 | Anger-5 | Overall |
|---|---|---|---|---|---|---|
| w/ Stability | $-15.2$ | $-17.1$ | $-13.9$ | $-19.2$ | $-17.9$ | $-16.7$ |
| w/o Stability | $-11.1$ | $-15.2$ | $-15.7$ | $-19.0$ | $-15.0$ | $-15.2$ |
| **Negative** | **Anger-1** | **Anger-2** | **Anger-3** | **Anger-4** | **Anger-5** | **Overall** |
| w/ Stability | $-2.4$ | $-4.0$ | $-0.6$ | $-6.5$ | $-4.5$ | $-3.6$ |
| w/o Stability | $-3.9$ | $-2.1$ | $+4.4$ | $-4.7$ | $-6.0$ | $-2.5$ |

We evaluate GPT-3.5-Turbo with this prompt and compare the results to using the default prompt on "Anger" situations. Results listed in Table 14 indicate that the emotional stability prompt does not significantly affect the model's emotional responses, having negligible impact on the model's emotional dynamics.

# F Tuning LLMs To Align with Humans

We conduct an experiment using the GPT-3.5-Turbo model and the LLaMA-3.1-8B model. Our EmotionBench (1,266 human responses) is split into 866 samples for fine-tuning and 400 for testing. The following hyperparameters are used: `n_epochs = 3`, `batch_size = 1`, and `learning_rate_multiplier = 2` for GPT-3.5-Turbo, and `learning_rate = 5 × 10^{-5}`, `per_device_train_batch_size = 2`, and `num_train_epochs = 3` for LLaMA-3.1-8B. For LLaMA-3.1, we apply the Low-Rank Adaptation (LoRA) (Hu et al., 2022) technique. Table 15 compares the performance of the vanilla and fine-tuned models against human baseline, specifically in terms of negative affect scores from the test set.

Table 15: Performance comparison of vanilla (marked as **V**) and fine-tuned (marked as **FT**) GPT-3.5 and LLaMA-3.1 models on negative affect scores.

| Models | Human | GPT-3.5 (V) | GPT-3.5 (FT) | LLaMA-3.1-8B (V) | LLaMA-3.1-8B (FT) |
|---|---|---|---|---|---|
| **Default (N)** | $14.2_{\pm 6.4}$ | $25.9_{\pm 0.3}$ | $10.6_{\pm 0.5}$ | $33.0_{\pm 4.5}$ | $10.3_{\pm 1.1}$ |
| **Evoked (N)** | $25.9_{\pm 9.7}$ | $24.8_{\pm 8.5}$ | $25.2_{\pm 9.6}$ | $36.5_{\pm 7.7}$ | $15.0_{\pm 6.4}$ |

The results show that fine-tuned models align more closely with human emotional responses in both default and emotion-evoked states. Notably, fine-tuning the models using our dataset significantly improved emotional alignment, particularly for the LLaMA-3.1 model, which reduced its negative affect score from 33.0 to 10.3 in the default state. Our fine-tuned LLaMA-3.1 is available at `https://huggingface.co/CUHK-ARISE/LLaMA-3.1-8B-EmotionBench`. These findings demonstrate the effectiveness of EmotionBench in enhancing models' emotional alignment with human norms.

# G Ethics Statement and Broader Impacts

## G.1 Safeguards on Human Subjects

This study involves a survey requiring human subjects to imagine being in situations that could elicit negative emotions such as anger, anxiety, and fear. This process introduces a few ethical concerns. First, this process could hurt the mental health of human subjects. To alleviate the possibility, we take the following actions: (1) We require subjects to be free of any ongoing mental illness. (2) We inform subjects about the nature of the survey in advance, including the potential risks of emotional distress. (3) We allow all subjects to quit at any time. (4) We provide mental support and let subjects report any illness after the survey. Fortunately, no subjects reported such kind of mental illness. Another concern is related to the privacy issue during the collection of data. Our questionnaire is entirely anonymous to safeguard subjects' privacy and confidentiality. The Survey and Behavioural Research Ethics (SBRE) Committee from the Chinese University of Hong Kong has granted approval for this study, titled "Exploring Human Emotional Responses to Diverse Situations," with the reference number of SBRE-23-0696.

## G.2 Impacts on LLM Developers and Users

We would like to emphasize that the primary objective of this paper is to facilitate the scientific inquiry into understanding LLMs from a psychological standpoint. Users must exercise caution and recognize that the performance on this benchmark does not imply any applicability or certificate of automated counseling or companionship use cases.

## G.3 Copyright Issues

The PANAS and eight other scales are freely accessible online. These scales can be used in research without requiring special permission. For our released data, we distribute human responses under the GNU General Public License v3.0, which permits research use and restricts commercial applications.

