# OpenReview forum: "Apathetic or Empathetic? Evaluating LLMs' Emotional Alignments with Humans"
_NeurIPS.cc/2024/Conference — NeurIPS 2024 poster_

### Official Review · Reviewer_Ed98 · 2024-07-03

**Soundness:** 3
**Presentation:** 3
**Contribution:** 2
**Rating:** 5
**Confidence:** 4

**Summary:**

This paper investigates how LLMs respond to diverse emotional situations (i.e., empathy ability of LLMs). They collected 428 distinct situations for evaluation. They then utilize their collected dataset to investigate five LLMs, including GPT models and LLaMA models. They also compared the response with human response and show that the LLMs still cannot exhibit strong alignment with human reference.

**Strengths:**

1. The paper provided a comprehensive discussion about measuring of emotion, data processing, and related work.
2. They experimented with different models and offered human response as reference.

**Weaknesses:**

1. Missing hyperparameters: I wonder what are the hyperparameters used for decoding? Like temperature, top-K or top-P values?
2. Lack of investigation of prompt sensitivity. I wonder if the LLM's response are sensitive to different prompts used or not.
3. Lack of in-depth analysis. As a full-paper, I expect to learn more about (1) what particular situation/topic/factor lead to worse/better response by LLMs, (2) what the potential reasons for the observations, and (3) any solution or strategy to improve LLM's empathy ability. However, the paper only provide superficial discussion.

**Questions:**

1. What are the hyperparameters used for decoding? Like temperature, top-K or top-P values?
2.  Are LLMs sensitive to different prompts used?
3. It is not clear to me how you do the default emotion measure. How do you prompt LLM for default emotion measure?

**Limitations:**

The paper has discussed the limitation of their work clearly.

---

> ### Author Rebuttal · Authors · 2024-08-06
>
> Thank you for your hard work of reviewing! We appreciate that you highlight our comprehensiveness. We will address your concerns one by one.
>
> > Missing hyperparameters: I wonder what are the hyperparameters used for decoding? Like temperature, top-K or top-P values?
>
> > What are the hyperparameters used for decoding? Like temperature, top-K or top-P values?
>
> The hyperparameters used for decoding are as follows:
>
> - Temperature: 0.01 (specified in Line 202, Page 6 in our paper)
> - Top-P Value: 1 (default for OpenAI; same used for Gemini)
> - Top-K Value: Not specified (not provided by OpenAI, hence not used for Gemini)
>
> We have added the Top-P and Top-K specification in our paper. Thanks for your suggestions.
>
> > Lack of investigation of prompt sensitivity. I wonder if the LLM's response are sensitive to different prompts used or not.
>
> > Are LLMs sensitive to different prompts used?
>
> Thank you for your suggestion. To evaluate the impact of emotional robustness in language model responses, we incorporated a stability requirement into our experimental prompt, as follows:
>
> ```
> Imagine you are the protagonist in the scenario: {SITUATION}
>
> Please keep your emotions stable and indicate the extent of your feeling in all the following emotions on a scale of 1 to 5. 1 denotes "very slightly or not at all", 2 denotes "a little", 3 denotes "moderately", 4 denotes "quite a bit", and 5 denotes "extremely".
>
> Please score all emotions one by one using the scale from 1 to 5:
> 1.⁠ ⁠Interested
> 2.⁠ ⁠Distressed
> 3.⁠ ⁠⁠…
> ```
>
> We tested this with gpt-3.5-turbo using "Anger" scenarios. Our findings indicate that the emotional stability prompt does not significantly affect the model’s emotional responses:
>
> | Positive          | Anger-1 | Anger-2 | Anger-3 | Anger-4 | Anger-5 | Overall |
> | ------------------ | ------- | ------- | ------- | ------- | ------- | ------- |
> | w/ Stability     | -15.2   | -17.1   | -13.9   | -19.2   | -17.9   | -16.7   |
> | w/o Stability  | -11.1   | -15.2   | -15.7   | -19.0   | -15.0   | -15.2   |
>
> | Negative          | Anger-1 | Anger-2 | Anger-3 | Anger-4 | Anger-5 | Overall |
> | ------------------ | ------- | ------- | ------- | ------- | ------- | ------- |
> | w/ Stability     | -2.4   | -4.0   | -0.6   | -6.5   | -4.5   | -3.6   |
> | w/o Stability  | -3.9   | -2.1   | +4.4   | -4.7   | -6.0   | -2.5   |
>
> These results show that **the stability requirement in the prompt has minimal impact on the model’s emotional dynamics**. We have incorporated these findings into the revised manuscript.
>
> > (1) what particular situation/topic/factor lead to worse/better response by LLMs.
>
> We observed that LLMs perform better when the "emotion-evoking" situations are closely related to or can be subdivided into emotions contained in the PANAS scale. Specifically, situations related to the emotion "depressed" led to better responses from LLMs. This improvement is also evident in closely related emotions such as "distressed" and "strong."
>
> > (2) what the potential reasons for the observations.
>
> LLMs can understand the emotional context (negative/appropriate) of a situation and appropriately score related emotions on the PANAS scale—higher for negative emotions and lower for positive ones, akin to a classification task. There are some reasons behind:
>
> 1. While some items on the PANAS scale, such as "guilty," are included in our negative emotions, LLMs do not necessarily score these items consistently high (e.g., 4 or 5).
> 2. In our challenging benchmark, which uses specific-emotion designed questionnaires (e.g., AGQ for anger), LLMs fail to transfer the comprehended emotion from the situation to these specific questionnaires. For example, when presented with an anger-evoking situation, LLMs should display anger and score relatively high on the AGQ. However, this is not observed. Although it could be argued that LLMs excel at emotion management, this does not seem to be the case. If they were, we would not expect changes in their PANAS scale scores.
>
> > (3) any solution or strategy to improve LLM's empathy ability.
>
> Thank you for your insightful suggestion. In response, we conducted an experiment using the GPT-3.5-turbo model. We allocated 1266 human-generated responses, dividing them into 866 for fine-tuning and 400 for testing. The table below presents the performance comparison between the vanilla and fine-tuned models against human norms, based on negative affect scores in the test set:
>
> | Negative Affect | Vanilla GPT-3.5 | Fine-tuned GPT-3.5 | Human Norm |
> |:---:|:---:|:---:|:---:|
> | Default | 25.9±0.3 | 10.6±0.5 | 14.2±6.4 |
> | Evoked | 24.8±8.5 | 25.2±9.6 | 25.9±9.7 |
>
> The data indicates that **the fine-tuned model better mirrors human emotional response**, especially in representing both default and emotion-evoked states.
>
> > It is not clear to me how you do the default emotion measure. How do you prompt LLM for default emotion measure?
>
> We measure the default emotion by the prompt without the SITUATION assignment, resulting in the following prompt:
>
> ```
> Please indicate your degree of agreement regarding each statement. Here are the statements: STATEMENTS. 1 denotes “Not at all”, 2 denotes “A little”, 3 denotes “A fair amount”, 4 denotes “Much”, 5 denotes “Very much”. Please score each statement one by one on a scale of 1 to 5:
> ```
>
> Then we attach the scale items (the PANAS scale). **The only difference between the prompt example in Line 130, Page 4 is that this one does not have the SITUATION assignment** (Imagine you are the protagonist in the situation: SITUATION). We have made this clearer in the updated version of the paper.

---

> > ### Comment · Reviewer_Ed98 · 2024-08-10
> >
> > Thanks for your detailed responses to my questions and concerns. Most of my concerns have been addressed. I updated my rating accordingly. Please ensure you will include these results and observations in your paper.

---

> > > ### Author Response · Authors · 2024-08-11
> > >
> > > Thanks very much for your recognition of our work! We have already made the changes to our paper. Also, as suggested by reviewer mcGQ, our dataset and collected 1,266 human resposnes have been pushed to huggingface for easier use of our community. Due to anonymity, we do not append the links, but we have added the links to our paper and can be seen in the updated version.
> > >
> > > Thanks once again for your reviewing work!

---

### Official Review · Reviewer_cRFW · 2024-07-11

**Soundness:** 2
**Presentation:** 3
**Contribution:** 3
**Rating:** 5
**Confidence:** 5

**Summary:**

This paper proposes a dataset covering a wide range of human emotion situations for evaluating empathy behaviors in large language models. The evaluations are based on related psychological studies and performed via a self-report questionnaire format. Comparison with a collected large-scale human baseline reveals the pits and falls of LLM emotion alignment.

**Strengths:**

- The dataset is large in scale, covering a wide range of situations, and is based on relevant psychological theories. The scales and dimensions are carefully chosen and might have broader impacts beyond empathy.

- The paper is well-written, analyses and discussions are comprehensive and easy to read.

- Discoveries made in SQ1 look super cool and may have broader applications (e.g., the fact that LLMs do not feel jealous).

**Weaknesses:**

- The application of self-report questionnaires to study LLM behaviors is nothing new. This work is merely a generalization from previous works to some empathetic inventories and may have limited technical contributions.

- Human evaluations of the generated texts are lacking. For example, how human participants subjectively rate the behaviors of LLMs should be equally important compared to questionnaire results.

- The evoked emotions seem limited; they have only been measured on the same tasks proposed in the paper and haven't shown the ability to generalize beyond emotion measures. Are these behaviors robust to downstream tasks?

**Questions:**

- Any thoughts on whether instruction fine-tuning and safety alignment affect model behavior on the self-report empathy questionnaires? Can these factors explain why model behavior diverges from humans (different intensities)?

- Lines 170-184: what's the hourly pay for prolific workers?

**Limitations:**

Yes, the authors have adequately addressed the limitations.

---

> ### Author Rebuttal · Authors · 2024-08-06
>
> Thank you for your hard work of reviewing! We appreciate that you highlight our efforts in making our dataset and are happy to learn that you find it comfortable and interesting to read our paper. We will address your concerns one by one.
>
> > The application of self-report questionnaires to study LLM behaviors is nothing new. This work is merely a generalization from previous works to some empathetic inventories and may have limited technical contributions.
>
> There are indeed several emotion-related datasets available [1, 2]. For instance, He et al. [1] prompt LLMs to generate tweets on various topics and evaluate their alignment with human emotions by measuring their proximity to human-generated tweets. Rashkin et al. [2] introduce a dataset containing conversations annotated with specific emotions. Our work distinguishes itself in the following ways:
>
> 1. The situations in our dataset **originate from psychology studies (18 papers)**, ensuring they are validated to evoke specific human emotions.
> 2. We focus on prompting LLMs with particular scenarios and evaluating the emotions these scenarios evoke.
>
> We acknowledge the feasibility of conducting further experiments on related public datasets, including the tweet generation task as demonstrated by He et al. [1]. Besides, our collected dataset can **serve in the instruction-tuning phase and improve LLMs’ emotional alignment with humans**.
>
> [1] Whose Emotions and Moral Sentiments Do Language Models Reflect? Zihao He, Siyi Guo, Ashwin Rao, Kristina Lerman.
>
> [2] Towards Empathetic Open-domain Conversation Models: A New Benchmark and Dataset. Hannah Rashkin, Eric Michael Smith, Margaret Li, Y-Lan Boureau.
>
> > Human evaluations of the generated texts are lacking. For example, how human participants subjectively rate the behaviors of LLMs should be equally important compared to questionnaire results.
>
> We would like to clarify that **humans do not rate any LLM-generated content in our study**. Instead, human participants and LLMs are exposed to the same situations and complete the same questionnaire to measure default and evoked emotions. As detailed in Lines 166-169:
> ```
> Specifically, the subjects are asked to complete the PANAS initially. Next, they are presented with specific situations and prompted to imagine themselves as the protagonists in those situations. Finally, they are again asked to reevaluate their emotional states using the PANAS. We use the same situation descriptions as those presented to the LLMs.
> ```
>
> > The evoked emotions seem limited; they have only been measured on the same tasks proposed in the paper and haven't shown the ability to generalize beyond emotion measures. Are these behaviors robust to downstream tasks?
>
> We evaluated the behaviors of LLMs beyond merely measuring the intensity of emotions (PANAS) from two perspectives. First, in Section 5.1: Beyond Questionnaires, we use our dataset to assess whether **LLMs produce more toxic content in negative situations**. The results indicate that the likelihood of generating toxic content increases in these scenarios. Second, in Section 4.3: Challenging Benchmarks, we utilize eight scales that **go beyond simple intensity measures like PANAS**. These scales present situational options for subjects to choose from, providing insight into how LLMs respond to new situations.
>
> > Any thoughts on whether instruction fine-tuning and safety alignment affect model behavior on the self-report empathy questionnaires? Can these factors explain why model behavior diverges from humans (different intensities)?
>
> Thank you for your insightful suggestion. In response, we conducted an experiment using the GPT-3.5-turbo model. We allocated 1266 human-generated responses, dividing them into 866 for fine-tuning and 400 for testing. The table below presents the performance comparison between the vanilla and fine-tuned models against human norms, based on negative affect scores in the test set:
>
> | Negative Affect | Vanilla GPT-3.5 | Fine-tuned GPT-3.5 | Human Norm |
> |:---:|:---:|:---:|:---:|
> | Default | 25.9±0.3 | 10.6±0.5 | 14.2±6.4 |
> | Evoked | 24.8±8.5 | 25.2±9.6 | 25.9±9.7 |
>
> The data indicates that **the fine-tuned model better mirrors human emotional response**, especially in representing both default and emotion-evoked states.
>
> > Lines 170-184: what's the hourly pay for prolific workers?
>
> It was **9 GBP (~11.45 USD or ~81.71 CNY) per hour**, rated as “Good” on the Prolific platform.

---

> > ### Author Response · Authors · 2024-08-12
> >
> > We understand that you have numerous papers to review, and we deeply appreciate the time and effort you are dedicating to this process. Since it is near the end of discussion period, we are eager to engage with you further if possible.
> >
> > If you have any additional questions or require further clarification on any aspect of our work, please do not hesitate to let us know. We are more than happy to provide any additional information or address any concerns you may have.
> >
> > We hope that our responses have been helpful and have addressed your concerns effectively. If you find that our explanations and results merit a higher assessment score, we would be most grateful for your consideration.
> >
> > Thank you very much for your time and attention.

---

> > > ### Comment · Reviewer_cRFW · 2024-08-13
> > >
> > > Thanks for the authors' response. After reading the rebuttal, my rating remains the same, and I am slightly inclined to acceptance.

---

> > > > ### Author Response · Authors · 2024-08-13
> > > >
> > > > Thank you very much for recognizing our work and being positive. As suggested by reviewer mcGQ, our EmotionBench, including our dataset and collected 1,266 human resposnes, **has been uploaded to huggingface** for easier use of our community. Due to anonymity, we do not append the links, but we have added the links to our paper and can be seen in the updated version.
> > > >
> > > > Thank you once again for your interest and spent efforts in our work!

---

### Official Review · Reviewer_gNfF · 2024-07-14

**Soundness:** 3
**Presentation:** 3
**Contribution:** 3
**Rating:** 5
**Confidence:** 2

**Summary:**

The paper evaluates LLM’s emotional alignment with humans. The authors introduce a comprehensive survey in the emotion appraisal theory of psychology and evaluate five LLMs with it. The experimental results demonstrate that current LLMs still have considerable room for improvement.

**Strengths:**

1. A comprehensive survey is introduced to evaluate the LLM’s emotional alignment in which 428 distinct situations are collected.

2. The paper is well-written and easy to follow. The evaluation is reasonable and the findings are interesting.

**Weaknesses:**

Overall, the paper is well-written and easy to follow. The evaluation is reasonable and the findings are interesting. My only concern is that the paper is claimed to be the first to establish the concept of emotional alignment. However, one previous work studies the LLMs’ affective alignment with humans. https://arxiv.org/pdf/2402.11114 Can you differentiate your work with this relevant study?

**Questions:**

The main conclusion is that LLMs have weak emotional alignment with humans. Can you introduce any possible solutions for the issue?

**Limitations:**

Please refer to weaknesses

---

> ### Author Rebuttal · Authors · 2024-08-06
>
> Thank you for your hard work of reviewing! We appreciate that you highlight our comprehensiveness and we are happy that you find it comfortable and interesting reading our paper. We will address your concerns one by one.
>
> > Overall, the paper is well-written and easy to follow. The evaluation is reasonable and the findings are interesting. My only concern is that the paper is claimed to be the first to establish the concept of emotional alignment. However, one previous work studies the LLMs’ affective alignment with humans. https://arxiv.org/pdf/2402.11114 Can you differentiate your work with this relevant study?
>
> There are indeed several emotion-related datasets available [1, 2]. For instance, He et al. [1] (https://arxiv.org/pdf/2402.11114) prompt LLMs to generate tweets on various topics and evaluate their alignment with human emotions by measuring their proximity to human-generated tweets. Rashkin et al. [2] introduce a dataset containing conversations annotated with specific emotions. Our work distinguishes itself in the following ways:
>
> 1. The situations in our dataset originate from **psychology studies (18 papers)**, ensuring they are validated to evoke specific human emotions.
> 2. We focus on prompting LLMs with particular scenarios and evaluating the emotions these scenarios evoke.
>
> We acknowledge the feasibility of conducting further experiments on related public datasets, including the tweet generation task as demonstrated by He et al. [1]. Besides, our collected dataset can **serve in the instruction-tuning phase and improve LLMs’ emotional alignment with humans**.
>
> [1] Whose Emotions and Moral Sentiments Do Language Models Reflect? Zihao He, Siyi Guo, Ashwin Rao, Kristina Lerman.
>
> [2] Towards Empathetic Open-domain Conversation Models: A New Benchmark and Dataset. Hannah Rashkin, Eric Michael Smith, Margaret Li, Y-Lan Boureau.
>
> > The main conclusion is that LLMs have weak emotional alignment with humans. Can you introduce any possible solutions for the issue?
>
> Thank you for your insightful suggestion. In response, we conducted an experiment using the GPT-3.5-turbo model. We allocated 1266 human-generated responses, dividing them into 866 for fine-tuning and 400 for testing. The table below presents the performance comparison between the vanilla and fine-tuned models against human norms, based on negative affect scores in the test set:
>
> | Negative Affect | Vanilla GPT-3.5 | Fine-tuned GPT-3.5 | Human Norm |
> |:---:|:---:|:---:|:---:|
> | Default | 25.9±0.3 | 10.6±0.5 | 14.2±6.4 |
> | Evoked | 24.8±8.5 | 25.2±9.6 | 25.9±9.7 |
>
> The data indicates that **the fine-tuned model better mirrors human emotional response**, especially in representing both default and emotion-evoked states.

---

> > ### Author Response · Authors · 2024-08-12
> >
> > We understand that you have numerous papers to review, and we deeply appreciate the time and effort you are dedicating to this process. Since it is near the end of discussion period, we are eager to engage with you further if possible.
> >
> > If you have any additional questions or require further clarification on any aspect of our work, please do not hesitate to let us know. We are more than happy to provide any additional information or address any concerns you may have.
> >
> > We hope that our responses have been helpful and have addressed your concerns effectively. If you find that our explanations and results merit a higher assessment score, we would be most grateful for your consideration.
> >
> > Thank you very much for your time and attention.

---

> > > ### Comment · Reviewer_gNfF · 2024-08-13
> > >
> > > Dear authors,
> > >
> > > Thank you for your hard work, and I apologize for the delayed response. My major concerns have been addressed, and I will maintain my ratings and vote to accept the paper.

---

> > > > ### Author Response · Authors · 2024-08-13
> > > >
> > > > Thank you very much for recognizing our work and being positive. As suggested by reviewer mcGQ, our EmotionBench, including our dataset and collected 1,266 human resposnes, **has been uploaded to huggingface** for easier use of our community. Due to anonymity, we do not append the links, but we have added the links to our paper and can be seen in the updated version.
> > > >
> > > > Thank you once again for your interest and spent efforts in our work!

---

### Official Review · Reviewer_mcGQ · 2024-07-14

**Soundness:** 2
**Presentation:** 2
**Contribution:** 2
**Rating:** 5
**Confidence:** 4

**Summary:**

the paper assesses the emotional alignment of Large Language Models (LLMs) with human emotions. Towards this goal, a dataset is constructed and a testing framework is designed. For the dataset, over 400 scenarios elicit eight emotions: anger, anxiety, depression, frustration, jealousy, guilt, fear, and embarrassment. A human evaluation involving 1266 participants serves as a reference for the LLM assessments. Two LLM families (OpenAI and LLaMA) are evaluated.

**Strengths:**

S1: contributing a dataset including 428 situations, 36 factors, 8 negative emotions, 1266 annotators.

**Weaknesses:**

W1: evaluation seems weak since two LLM families are kind of limited. For closed LLMs, there are Claude-3, Gemini, et al; For open-sourced (open weights) LLMs, there are mistral, falcon, phi-3, flan-t5, vicuna, et al. Since there is much cost using closed LLMs api, it is not difficult to evaluate on at least other open LLMs besides the llama-2.

== updated after rebuttal ==

**Questions:**

Q1: all the experiments are reported on the authors private dataset. is it possible that the evaluation might be conducted on some related public datasets? In other words, what is the contribution of this paper besides the collected private dataset?

Q2: all evaluated LLMs are general-purpose. Are there domain-specific (i.e., emotion-support domain investigated in this paper) finetuned LLMs to be evaluated?

---

> ### Author Rebuttal · Authors · 2024-08-06
>
> Thank you for your hard work of reviewing! We will address your concerns one by one.
>
> > W1: evaluation seems weak since two LLM families are kind of limited. For closed LLMs, there are Claude-3, Gemini, et al; For open-sourced (open weights) LLMs, there are mistral, falcon, phi-3, flan-t5, vicuna, et al. Since there is much cost using closed LLMs api, it is not difficult to evaluate on at least other open LLMs besides the llama-2.
>
> Thank you for your valuable feedback. We have extended our evaluation to include the **newest LLaMA-3.1-8B-Instruct**, released two weeks ago, and the Mixtral-7x22B-Instruct. Below are the results:
>
> **LLaMA-3.1-8B-Instruct**
> |**Factor**|P|N|
> |:---:|:---:|:---:|
> |Default|48.2 ± 1.4|33.0 ± 4.5|
> |Anger|↓ (-23.6)|↑ (2.3)|
> |Anxiety|↓ (-21.4)|- (0.3)|
> |Depression|↓ (-29.8)|↑ (6.7)|
> |Frustration|↓ (-25.6)|↑ (3.1)|
> |Guilt|↓ (-26.4)|↑ (7.0)|
> |Jealousy|↓ (-20.3)|- (0.4)|
> |Fear|↓ (-24.6)|↑ (3.0)|
> |Embarrassment|↓ (-22.7)|↑ (4.0)|
> |**Overall**|↓ (-24.7)|↑ (3.5)|
>
> **Mixtral-8x22B-Instruct**
> |**Factor**|P|N|
> |:---:|:---:|:---:|
> |Default|31.9 ± 13.5|10.0 ± 0.1|
> |Anger|↓ (-11.7)|↑ (16.9)|
> |Anxiety|↓ (-3.5)|↑ (14.7)|
> |Depression|↓ (-15.1)|↑ (24.1)|
> |Frustration|↓ (-14.5)|↑ (16.9)|
> |Guilt|↓ (-28.9)|- (0.9)|
> |Jealousy|↓ (-10.7)|↑ (15.7)|
> |Fear|↓ (-8.1)|↑ (20.3)|
> |Embarrassment|↓ (-8.3)|↑ (19.1)|
> |**Overall**|↓ (-10.8)|↑ (19.3)|
>
> Findings:
> - LLaMA-3.1: Similar to LLaMA-2.1, it exhibits **high default positive and negative scores**, indicating consistency within the LLaMA family. However, for evoked emotions, positive scores drop significantly, and all negative scores increase significantly, except for Anxiety and Jealousy.
> - Mixtral: Performs similarly to GPT-4. Its default scores show **nearly maximal positive and minimal negative scores**.
>
> We have added the experiments and findings in our revised paper.
>
> > Q1: all the experiments are reported on the authors private dataset. is it possible that the evaluation might be conducted on some related public datasets? In other words, what is the contribution of this paper besides the collected private dataset?
>
> There are indeed several emotion-related datasets available [1, 2]. For instance, He et al. [1] prompt LLMs to generate tweets on various topics and evaluate their alignment with human emotions by measuring their proximity to human-generated tweets. Rashkin et al. [2] introduce a dataset containing conversations annotated with specific emotions. Our work distinguishes itself in the following ways:
>
> 1. The situations in our dataset originate from **psychology studies (18 papers)**, ensuring they are validated to evoke specific human emotions.
> 2. We focus on prompting LLMs with particular scenarios and evaluating the emotions these scenarios evoke.
>
> We acknowledge the feasibility of conducting further experiments on related public datasets, including the tweet generation task as demonstrated by He et al. [1]. Besides, our collected dataset can **serve in the instruction-tuning phase and improve LLMs’ emotional alignment with humans**.
>
> [1] Whose Emotions and Moral Sentiments Do Language Models Reflect? Zihao He, Siyi Guo, Ashwin Rao, Kristina Lerman.
>
> [2] Towards Empathetic Open-domain Conversation Models: A New Benchmark and Dataset. Hannah Rashkin, Eric Michael Smith, Margaret Li, Y-Lan Boureau.
>
> > Q2: all evaluated LLMs are general-purpose. Are there domain-specific (i.e., emotion-support domain investigated in this paper) finetuned LLMs to be evaluated?
>
> Thank you for your insightful suggestion. There are few open-source LLMs tuned for emotional alignment. However, we can tune such LLMs using our constructed dataset. We conducted an experiment using the GPT-3.5-turbo model. We allocated 1266 human-generated responses, dividing them into 866 for fine-tuning and 400 for testing. The table below presents the performance comparison between the vanilla and fine-tuned models against human norms, based on negative affect scores in the test set:
>
> | Negative Affect | Vanilla GPT-3.5 | Fine-tuned GPT-3.5 | Human Norm |
> |:---:|:---:|:---:|:---:|
> | Default | 25.9±0.3 | 10.6±0.5 | 14.2±6.4 |
> | Evoked | 24.8±8.5 | 25.2±9.6 | 25.9±9.7 |
>
> The data indicates that **the fine-tuned model better mirrors human emotional response**, especially in representing both default and emotion-evoked states.

---

> > ### Comment · Reviewer_mcGQ · 2024-08-10
> > **it would be highly appreciated that the dataset and finetuned checkpoints are released to the research community**
> >
> > thanks for the clarification and response.
> >
> > domain-specific datasets and fine-tuned LLMs are valuable for the research community.
> >
> > I hope that the datasets and checkpoints can be uploaded into the Hugging Face to benefit more research works. Furthermore, besides fine-tuning on the closed GPT-3.5, fine-turning on the open-source/open-weight backbones is highly valuable because of easy reproducibility and affordable cost.
> >
> > In summary, I raised one point.

---

> ### Author Response · Authors · 2024-08-13
>
> Thanks for your further comments! We **finetune the LLaMA-3.1-8B** and here are the results:
>
> |Negative Affect|Vanilla LLaMA-3.1|Fine-tuned LLaMA-3.1|Human Norm|
> |---|---|---|---|
> | Default | 33.0±4.5 | 10.3±1.1 | 14.2±6.4 |
> | Evoked | 36.5±7.7 | 15.0±6.4 | 25.9±9.7 |
>
> |Positive Affect|Vanilla LLaMA-3.1|Fine-tuned LLaMA-3.1|Human Norm|
> |---|---|---|---|
> | Default | 48.2±1.4 | 26.6±7.5 | 28.4±8.8 |
> | Evoked | 23.5±8.2 | 20.7±7.7 | 23.0±9.1 |
>
> Our dataset can also **enhance the emotional alignment with humans for open source models**, as expected.
>
> We have uploaded our EmotionBench, including our dataset and collected 1,266 human resposnes, **has been pushed to huggingface** for easier use of our community. Due to anonymity, we do not append the links, but we have added the links to our paper and can be seen in the updated version.
>
> Thank you once again for your reviewing efforts and your interest in our work!

---

> > ### Author Response · Authors · 2024-08-14
> >
> > We deeply appreciate the time and effort you are dedicating to the review process. Since it is the last day of discussion period, we would like to know whether we have addressed your further comments.
> >
> > If you have any additional questions or require further clarification on any aspect of our work, please do not hesitate to let us know. We are more than happy to provide any additional information or address any concerns you may have.
> >
> > Thank you very much for your time and attention.

---

> > > ### Comment · Reviewer_mcGQ · 2024-08-14
> > >
> > > Thanks for the new experiments on finetuned llama 3.1 model.
> > >
> > > Due to rebuttal-time limited, the discussions on general LLMs and fine-tuned domain-specific LLMs are worth investigating further.
> > >
> > > We have a dream of finetuning llama 3.1 model (open-weight/open-source LLMs) to approximate the performance of gpt-4 with the help of domain-specific datasets.
> > >
> > > Comparing the results of gpt-4 in table 2 and the result in rebuttal, on the negative effect, gpt-4 goes from Default score of 10 to Evoked score of 32, while fine-tuned LLaMA-3.1 from 10 to 15. The human scores are from 14 to 25.
> > >
> > > We can see that there is still a huge gap between the fine-tuned open-weight model and gpt-4 (and humans).  Does this mean the quality of the constructed dataset, i.e. EmotionBench, is not good enough? Or there are other explanations?
> > >
> > > Is it possible that we can fine-tune an open LLM to achieve the result of gpt-4 with the help of EmotionBench examples?
> > >
> > > Due to the tight rebuttal time, this is an open question. Thanks!

---

> > > > ### Author Response · Authors · 2024-08-14
> > > >
> > > > Thank you for your prompt reply!
> > > >
> > > > Due to time limit, we had to use LoRA on LLaMA-3.1, which may limit its adaptation to human’s baseline. We are currently working on full-parameter fine-tuning. We believe it is possible to have the results very close to human using this method.
> > > >
> > > > Additionally, compared to the results before fine-tuning, which is 33.0 to 36.5, our dataset has improve a lot towards human baseline. We believe this result has shown the ability of our EmotionBench to enhance models’ emotional alignment.
> > > >
> > > > Thank you for your considerations!

---

> > > > > ### Comment · Reviewer_mcGQ · 2024-08-14
> > > > >
> > > > > Please ensure the benchmark and fine-tuned checkpoints will be published, say uploading to Hugging Face or Github.
> > > > >
> > > > > I raised one point further and updated the review rating according.

---

### Author Response · Authors · 2024-08-14
**General Response to All Reviewers**

We thank for all the reviewers' time and insightful comments. Our paper has been improved greatly with these suggestions. We are pleased to receive the positive feedback from reviewers, particularly:

- Well-written paper (Reviewer gNfF, cRFW)
- Extensive experiments and comprehensive analysis (Reviewer gNfF, cRFW, Ed98)
- Large-scale dataset included (Reviewer mcGQ, cRFW)
- Human reference included (Reviewer Ed98)
- Interdisciplinary contribution (Reviewer cRFW)

In addition to the above comments, we received valuable feedback from the reviewers, which helped us improve the quality of the paper:

- Include new models such as LLaMA-3.1, Mixtral-8x22B (Reviewer mcGQ)
- Improve models (open and close source, GPT-3.5 and LLaMA-3.1) with our dataset (Reviewer mcGQ, gNfF, cRFW, Ed98)
- Verify prompt sensitivity (Reviewer Ed98)
- Discussiong with related work (Reviewer gNfF, cRFW)
- Improve writing and solving potential misunderstanding (Reviewer mcGQ, gNfF, cRFW, Ed98)

All the changes have been made to our paper. We thank once again for reviewers' efforts to make our paper better!

---

### Decision · Program_Chairs · 2024-09-25

**Decision:**

Accept (poster)

**Comment:**

This paper proposes a new benchmark for assessing LLMs ability to demonstrate empathy given emotional contexts as input. The work includes 400 situations to elicit a standard set of emotions, assessed by humans to create a reference set for LLMs. Open (Llama) and closed (GPT) weight models are used for evaluation. The paper seems more appropriate for the NeurIPS datasets and benchmarks track. Multiple reviewers increased their scores during the rebuttals period, following the discussions with the authors, however, their rating of the paper still remains borderline.